# What Determines Utility of International Currencies?

**Eiji Ogawa** [1,2,*] **and Makoto Muto** [1]

[1]   Graduate School of Business Administration, Hitotsubashi University, Tokyo 186-8601, Japan;
      cd162004@g.hit-u.ac.jp
[2]   Research Institute of Economy, Trade and Industry (RIETI), Tokyo 100-8901, Japan
*   Correspondence: eiji.ogawa@r.hit-u.ac.jp

**Abstract:** In previous studies, we estimated a time series of coefficients on five international currencies (the US dollar, the euro, the Japanese yen, the British pound, and the Swiss franc) in a utility function. We call the coefficients utilities of international currencies. The time series show that the utility of the US dollar as an international currency has remained in the first position in the changing international monetary system despite of the fact that the euro was created as a single common currency for European countries. On one hand, the utility of the Japanese yen has been declining as an international currency. In this paper, we investigate what determines the utility of international currencies. We use a dynamic panel data model to analyze the issue with Generalized Method of Moments (GMM). Specifically, liquidity shortage in terms of an international currency means that it is inconvenient for economic agents to use the relevant currency for international economic transactions. In other words, liquidity shortages might reduce the utility of an international currency. In this analysis we focus on liquidity premium which represents a liquidity shortage in terms of an international currency. Our empirical results showed not only inertia in terms of change but also the impact of a liquidity shortage in an international currency on the utility of the relevant international currency.

**Keywords:** utility of international currency; inertia; liquidity risk premium; US dollar; Japanese yen

## 1. Introduction

The United States (US) dollar had been as a rule a key currency in the Bretton Woods international monetary system. The monetary authority of the United States fixed the US dollar to gold while the monetary authorities of other countries fixed their home currencies to the US dollar under the Bretton Woods system. It could keep stability of exchange rates among the currencies in the world economy. However, the Bretton Woods system was collapsed in 1971 because the monetary authority of the United States could not keep a value of the US dollar against gold to stop convertibility of the US dollar to gold. Afterwards, a position of the US dollar as a key currency has been still kept in the current international monetary system even though we have no longer the rule under which we have to use the US dollar as a key currency. The phenomenon is called as inertia of a key currency.

Given that a key currency is chosen for economic reasons which include costs and benefits of an international currency, comparison in costs and benefits of international currencies determines a key currency in the current international monetary policy. Also, inertia of a key currency should be related with inertia of costs and/or benefits of holding an international currency. The costs of holding an international currency are related with its depreciation that caused by inflation in the relevant country. On one hand, the benefits of holding an international currency are caused by utility of holding it.

In a Sidrauski (1967)-type of money-in-the-utility model (Calvo 1981, 1985; Obstfeld 1981; Blanchard and Fischer 1989), real balances of money as well as consumption are supposed as explanatory variables

in a utility function. We can use the money-in-the-utility model to analyze costs and benefits of holding international currencies. Ogawa and Muto (2017a, 2017b) used expected inflation rates and Bank for International Settlements (BIS) data on total of domestic currency denominated debt and foreign currency denominated debt of the euro currency market to estimate time series of coefficients on five international currencies (the US dollar, the euro, the Japanese yen, the British pound, and the Swiss franc) in a utility function. We call the coefficients utility of an international currency. The time series show that utility of the US dollar as an international currency has kept at the first position even though the euro was introduced into some of the European Union (EU) states while it increased utility of the euro as an international currency. On one hand, utility of the Japanese yen has been declining as an international currency. Since 1973, although the US dollar is downward trend, it has kept the key currency in the changing international monetary system. This is probably because the US dollar has reduced the store of value function but maintained the medium of exchange function. Utility of the international currency means relative contribution of holding an international currency through such functions of international currency. Therefore, we can estimate relative position of international currency from a value of utility of the international currency.

In this paper, we have an objective to investigate what determines utility of the international currencies. We use a dynamic panel data model to analyze the issue with Generalized Method of Moments (GMM). Specifically, liquidity shortage in terms of an international currency means that it is inconvenient for economic agents to use the relevant currency for international economic transactions. In other words, the liquidity shortage might reduce utility of an international currency. In this analysis we focus on liquidity premium which represents liquidity shortage in terms of an international currency. We make empirical analysis of whether liquidity risk premium in an international currency affects utility of the relevant international currency. For example, if the currency authority aims to internationalize its home currency, results of this analysis will be useful for which variables should be focused.

We obtain the following results from the empirical study. Firstly, change in utility of the currency in the previous period has significantly a positive effect on the change of utility of the currency in the current period. This suggests that utility of the currency tends to fluctuate in the same direction as the change in the previous period. For example, if the utility of the currency declines, we assumed that the currency is less likely to be used than in the previous period, which will continue in the next period. Secondly, the change of liquidity risk premium has a significantly negative effect on the change of utility of the currency. This suggests that liquidity shortage reduce the utility of the international currency. Thirdly, the change of capital flow share has significantly a positive effect on the change of utility of the currency. This suggests that changes in economic scale, specifically capital flow, affect the utility of the international currency.

In the next section, we describe related literatures. In the third section, we explain our theoretical model in terms of utility of an international currency. In the fourth section, we explain empirical model for analyzing determinants of utility of an international currency. In the fifth section, we explain data used for the analysis and calculation method. In the sixth section, we discuss hypothesis of estimated coefficients and influence of each variable on utility of an international currency. In the seventh section, we show results of dynamic panel analysis. Finally, we conclude our empirical analysis.

## 2. Related Literature

Krugman (1984) adopted three functions of money as a medium of exchange, a unit of account, and a store of value to consider six roles of an international currency for both private and official sectors. According to his definition, it is used as a medium of exchange in private international economic transactions ("vehicle" currency or settlement currency), while it is transacted by monetary authorities in order to intervene in foreign exchange markets ("intervention" currency). Private sector makes trade contracts which are denominated in terms of a currency ("invoice" currency). Monetary authorities set par values for exchange rates which are stated in terms of a currency ("peg" currency).

Private sector holds liquidity dollar denominated assets ("banking" role) as a store of value. Also, monetary authorities hold a currency as an international reserve ("reserve" currency) which is related with a store of value. Matsuyama et al. (1993) and Trejos and Wright (1996) used a search theory to investigate a role of international currency as a medium of exchange. Moreover, Kannan (2009) focused on the benefits arising from terms of trade as well as traditional seigniorage and presented models on the benefits of international currency. It was showed that the benefits arising from terms of trade are important.

Related studies focused on one of the functions of an international currency to investigate roles of a currency as an international currency and international monetary system with the US dollar as a key currency. For example, Chinn and Frankel (2007, 2008) focused on a role as international reserve currency. Eichengreen et al. (2016b) focused on a role of international reserve currency to investigate whether it has changed in the determinants of the currency composition of international reserves in before and after the collapse of the Bretton Woods regime. Goldberg and Tille (2008) analyzed the US dollar and other currencies as an invoice currency in international economic transactions. Ito et al. (2013) conducted a questionnaire survey on the choice of invoice currency with all Japanese manufacturing firms listed in the Tokyo Stock Exchange to show that the Japanese firms use the Japanese yen second to an importing country currency as invoice currency in exporting products to the US and Europe, while the Japanese yen is the first used in exporting them to Asia.

Catão and Terrones (2016) and Honohan (2008) focused on the dollarization of financial systems in emerging market economies. Especially, Catão and Terrones (2016) pointed out a broad global trend towards financial sector de-dollarization from the early 2000s to the eve of the global financial crisis. Kamps (2006) focused on the euro to investigate the decision on invoice currency in international trade. An analytical result is that economic agents in EU states played a role in determining the euro as an invoice currency. However, it was suggested that the US dollar is dominant as an invoice currency compared with the euro. ECB (European Central Bank 2015) reported increasing roles of the euro as an international currency in terms of each of the three functions in the international reserve, international trade, and financial markets.

Eichengreen et al. (2016a) conducted an empirical analysis on the international currency used in the settlement currency in the oil market using data from the 1930s to 1950s. Although the US dollar is said to be strongly dominant in the oil market, they showed that currencies other than dollars were used as the settlement currency to some extent in European countries and countries with stable currencies. These results showed that multiple international currencies were served as a means of settlement even in markets of such homogeneous goods as oil. They suggested that a transition from a dollar-based system to a multipolar system is not impossible.

## 3. Utility of International Currency

### 3.1. Estimation Equation of Utility of International Currency

Ogawa and Muto (2017a, 2017b) estimated a coefficient on each of international currencies in the utility function or utility of international currencies, given that economic agents make dynamic optimization of utility in a money-in-the-utility function while they faced depreciation of international currency holdings. They have optimal holdings of an international currency by comparing benefits or utility from holding it with costs or depreciation of holding it. We can derive invisible utility of an international currency as a function of visible economic variables which include holdings of an international currency and its depreciation. We can obtain an estimate of utility of an international currency $i$ ($\gamma_t^i$) according to the following estimation equation[1]:

---

[1]    See Appendix A for derivation of Equation (1). We suppose that $\gamma$ might change over time because we have an important objective to investigate what factors influence utility of the currency $\gamma$ during the analytical period though it seems to be stable as an exogenous.

$$\gamma_t^i = \cfrac{1}{1 + \left(\cfrac{1}{\phi_t^i} - 1\right)\cfrac{\pi_t^O + \bar{r}}{\pi_t^i + \bar{r}}} \tag{1}$$

where $\phi_t^i$: share of holdings of an international currency $i$, $\pi_t^i$: expected inflation (or depreciation) rate of country $i$, $\pi_t^O$: expected inflation (or depreciation) rate of the other countries, $\bar{r}$: real interest rate. Assumptions of both purchasing power parity and uncovered interest rate parity make real interest rates are equal to each other in the world.

In our previous study, we assumed real interest rates are 1.5%, 2.0%, 2.5%, and 3.0%.[2] In addition, there is also utility of an international currency calculated using the nominal interest rate as well as the expected inflation rate plus the real interest rate. However, the nominal interest rate has periods of zero-bound level. Moreover, it is considered that the nominal interest rate has a strong relationship with a liquidity risk premium. Therefore, in this analysis, utility of the international currency calculated using real interest rate was used.

### 3.2. Data for Estimating Utility of International Currencies

We should use data on shares of the international currencies according to the theoretical money-in-the-utility model in which they are regarded as real balances of international currencies. However, it is difficult to obtain data on the real balance of international currencies which include international currencies held by private sector in the world economy. Instead, we use BIS data on total of domestic currency denominated debt and foreign currency denominated debt of the euro currency market. The data are obtained from a BIS website.

The expected inflation rates are calculated rate of change between actual price level and expected price level estimated under the assumption that the price level of each period follows ARIMA (p, d, q) process[3]. We use monthly data on the price level for the last twenty-five years to estimate an ARIMA model. The Augmented Dickey–Fuller test is used to unit root test. The BIC is used for lag selection. The estimated ARIMA model is used to predict a price level of three periods ahead. Thus, we use the actual price level and the predicted price level of three periods ahead to calculate the expected inflation rate. Consumer price index (CPI) data are used as the price level. The data are obtained from the OECD website.

The expected inflation rate in the euro zone is a weighted average of the expected inflation rate in the original euro zone countries. The euro zone includes Austria, Belgium, Finland, France, Germany, Ireland, Italy, Luxembourg, Netherlands, Portugal, and Spain. A weight in calculating a weighted average of the expected inflation rate is based on their GDP share among the countries. The data were obtained from the *International Financial Statistics (IFS)* of International Monetary Fund (IMF) website.

### 3.3. Movements of Utility of International Currencies

We use Equation (1) to calculate utility of international currency in each period. Figure 1a–d show time series of utility of four international currencies. Throughout a whole period, changes in utility of the US dollar, the euro, the Japanese yen and the British pound are fluctuating around 0.5, 0.35, 0.03 and 0.08, respectively.

---

[2]  An arithmetic average of real economic growth rates compared to the same quarter of previous year among the three countries and the region (the United States, the euro zone, Japan, and the United Kingdom) was about 1.1% from 2006Q3 to 2017Q4. However, if we exclude a period of 2008Q2 to 2010Q1 where the growth rate has greatly declined due to the global financial crisis, it was about 1.8%. Given the real economic growth rates, our setting the values as a real interest rate seem to be reasonable. The real economic growth rate data obtained from the OECD website.

[3]  We used a method of Fama and Gibbons (1984) to estimate expected inflation rates. However, a sample period is much shorter than that by using the ARIMA model due to data constraints if we use the method. In addition, we could not use it because expected inflation rate of TIPS and survey data was only long-term expectation data, and Japan's TIPS data was a small sample. For those reasons, we choose to use the ARIMA model using CPI.

We can find that utility of the US dollar sharply decreased while the other currencies increased in 2008Q3. On the other hand, utility of other international currencies has increased. In particular, utility of the euro has greatly increased. Causes of this sharp change are considered as follows.

Inflation rate in the United States is relatively decreased compared with the other countries and region. At the Lehman Brothers bankruptcy in September 2008, housing price and rents in the United States sharp declined. Accordingly, CPI in the United States, which is greatly affected by housing price and rent, dropped in the period. Figure 2a–d show movements of CPI and expected price levels estimated from CPI. From the figures, from 2008Q3 to 2008Q4, the CPI of the United States is relatively lower than in other countries. Figure 3 shows movements of the four countries' expected inflation rate. From this figure, the expected inflation rate in the United States made larger decrease than the others in 2008Q3.

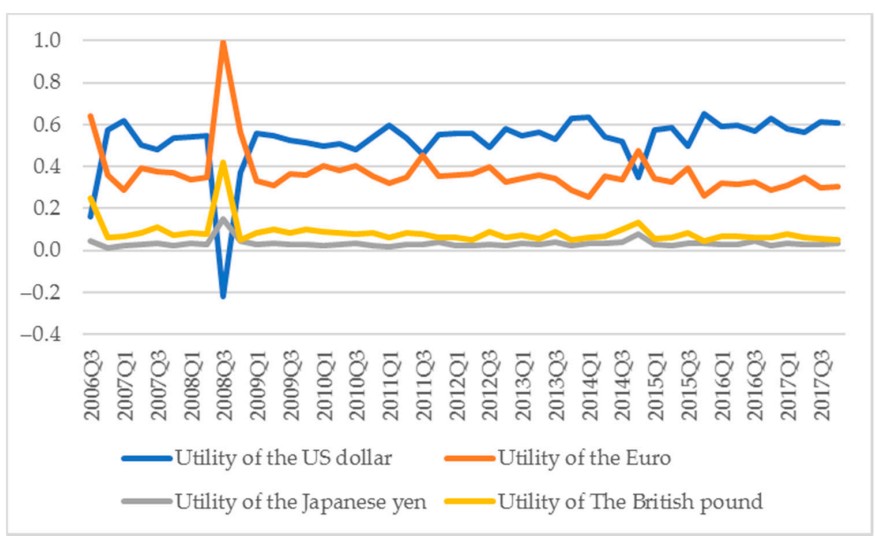

(**a**)

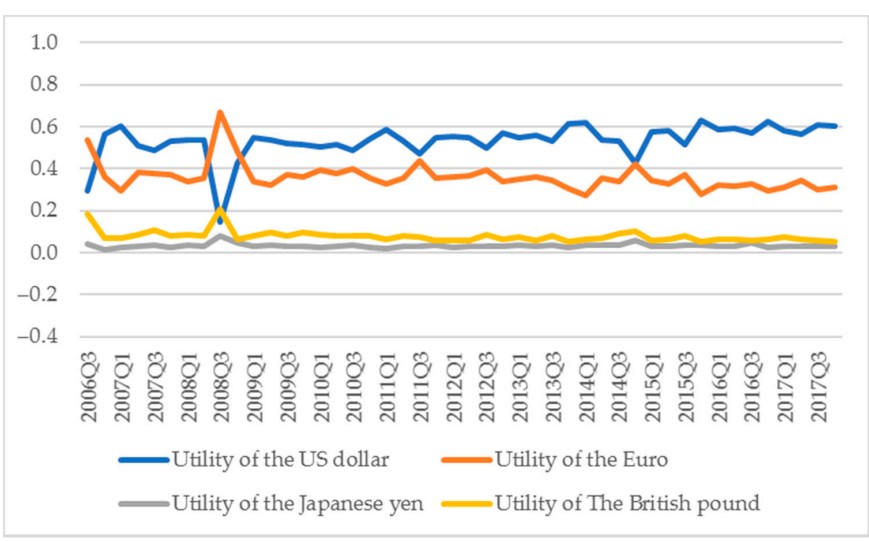

(**b**)

**Figure 1.** *Cont.*

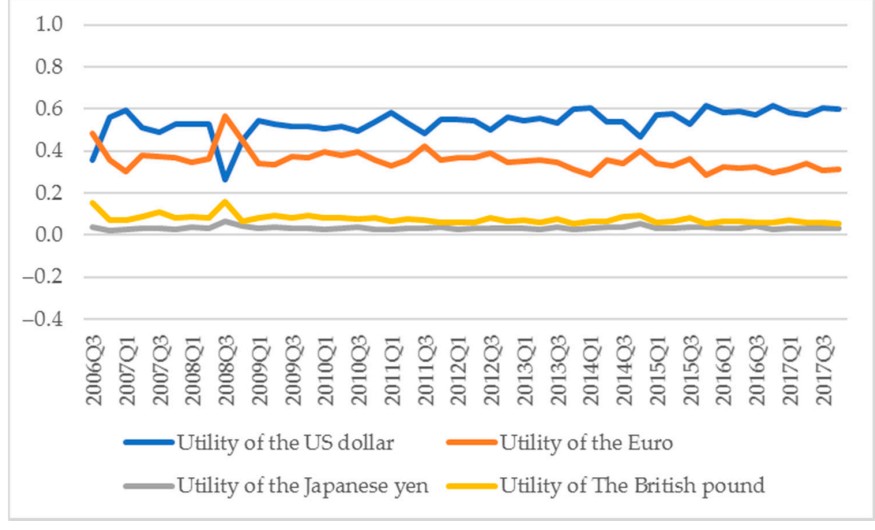

(**c**)

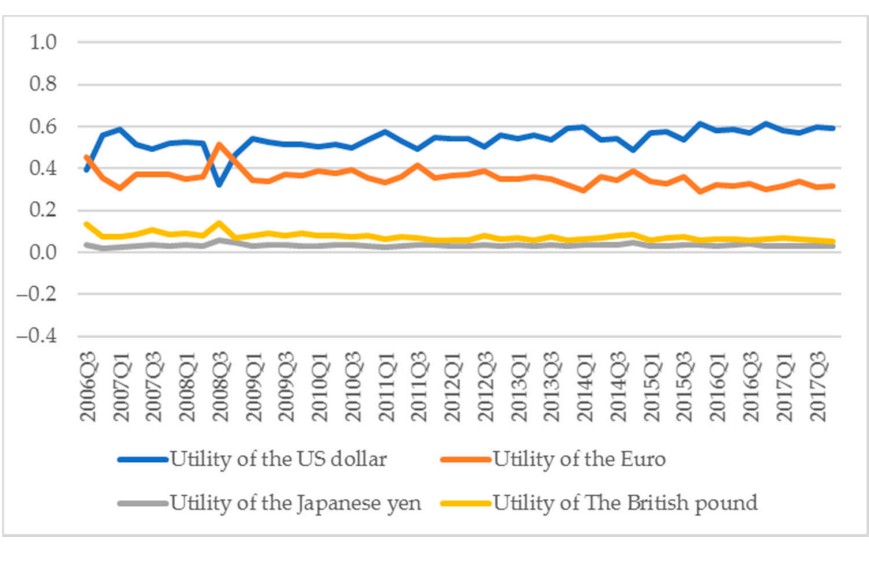

(**d**)

**Figure 1.** (**a**) Utility of international currencies (real interest rate = 1.5%). Notes: The four lines represent time series of estimated coefficients on four international currencies (the US dollar, the euro, the Japanese yen, and the British pound) in a money-in-the-utility function. The coefficients were estimated from share of holdings of an international currency and expected inflation rates with a real interest rate supposed to be 1.5%. We used BIS data on total of domestic currency denominated debt and foreign currency denominated debt of the euro currency market as the share of holdings of an international currency. The expected inflation rates are calculated rate of change of actual CPI level and expected CPI level estimated under the assumption that the price level of each period follows ARIMA (p, d, q) process. (**b**) Utility of international currencies (real interest rate = 2.0%). Notes: The four lines represent time series of estimated coefficients on four international currencies (the US dollar, the euro, the Japanese yen, and the British pound) in a money-in-the-utility function. The coefficients were estimated from share of holdings of an international currency and expected inflation rates with a real interest rate supposed to be 2.0%. We used BIS data on total of domestic currency denominated debt and foreign currency denominated debt of the euro currency market as the share of holdings of an international currency. The expected inflation rates are calculated rate of change of actual CPI level and

expected CPI level estimated under the assumption that the price level of each period follows ARIMA (p, d, q) process. (**c**) Utility of international currencies (real interest rate = 2.5%). Notes: The four lines represent time series of estimated coefficients on four international currencies (the US dollar, the euro, the Japanese yen, and the British pound) in a money-in-the-utility function. The coefficients were estimated from share of holdings of an international currency and expected inflation rates with a real interest rate supposed to be 2.5%. We used BIS data on total of domestic currency denominated debt and foreign currency denominated debt of the euro currency market as the share of holdings of an international currency. The expected inflation rates are calculated rate of change of actual CPI level and expected CPI level estimated under the assumption that the price level of each period follows ARIMA (p, d, q) process. (**d**) Utility of international currencies (real interest rate = 3.0%). Notes: The four lines represent time series of estimated coefficients on four international currencies (the US dollar, the euro, the Japanese yen, and the British pound) in a money-in-the-utility function. The coefficients were estimated from share of holdings of an international currency and expected inflation rates with a real interest rate supposed to be 3.0%. We used BIS data on total of domestic currency denominated debt and foreign currency denominated debt of the euro currency market as the share of holdings of an international currency. The expected inflation rates are calculated rate of change of actual CPI level and expected CPI level estimated under the assumption that the price level of each period follows ARIMA (p, d, q) process.

Next, the share of holdings of US dollar did not decrease although the inflation rate relatively decreased. In general, when an inflation rate decreases, a share of holdings of a currency will increase if utility of the currency does not change. On the other hand, if the share of holdings of a currency does not change, utility of the relevant currency decreases when the inflation rate decreases. Figure 4 shows movements in shares of total of domestic currency denominated debt and foreign currency denominated debt of the euro currency market. Figure 5 shows rate of change in shares of total of domestic currency denominated debt and foreign currency denominated debt of the euro currency market. From these figures, change in US dollar share from 2008Q3 to 2008Q4 is the second smallest. Moreover, as mentioned above, inflation rate in the United States at this time has decreased relatively. Therefore, utility of the US dollar decreased, given that the share was not changed and that the inflation rate relatively decreased.

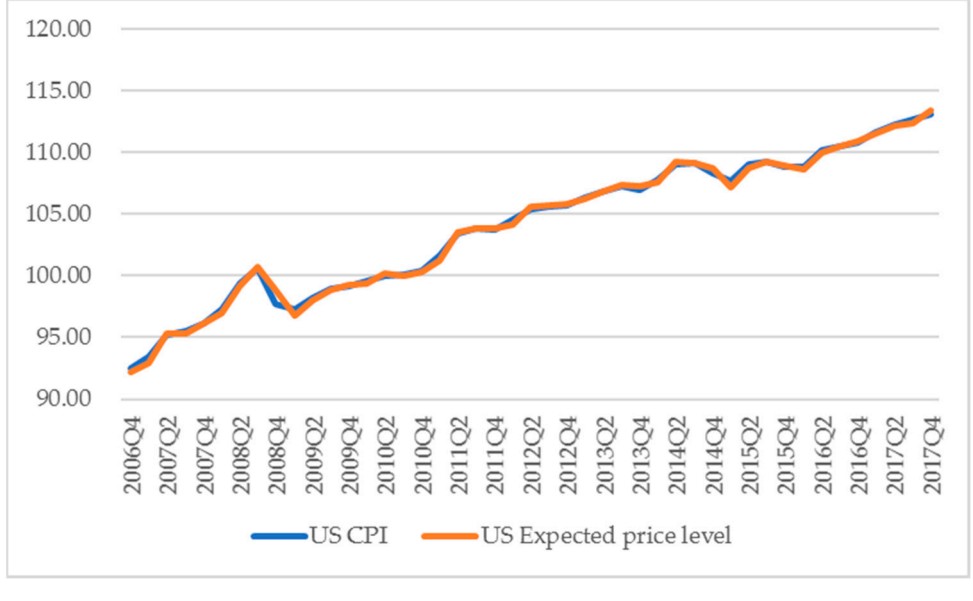

(**a**)

**Figure 2.** *Cont.*

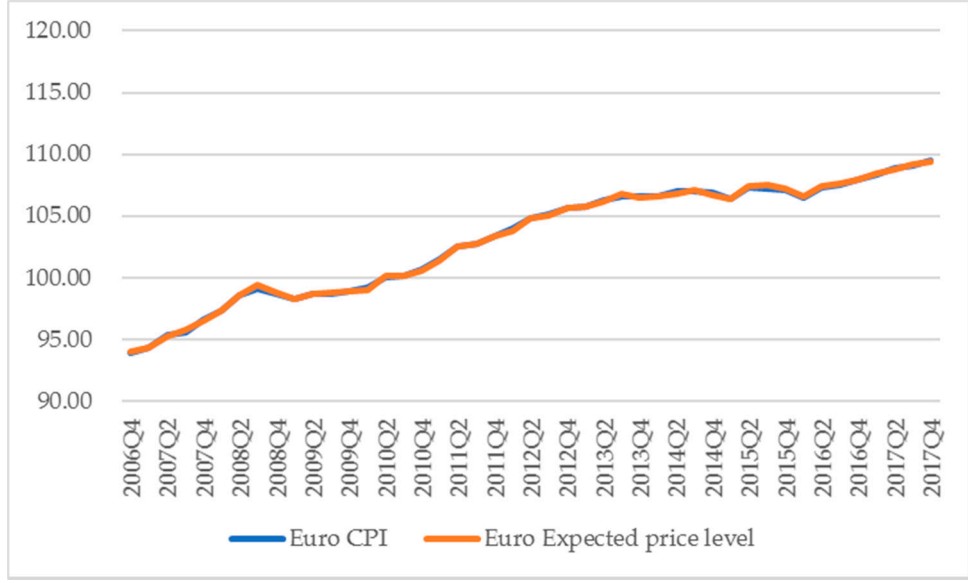

(**b**)

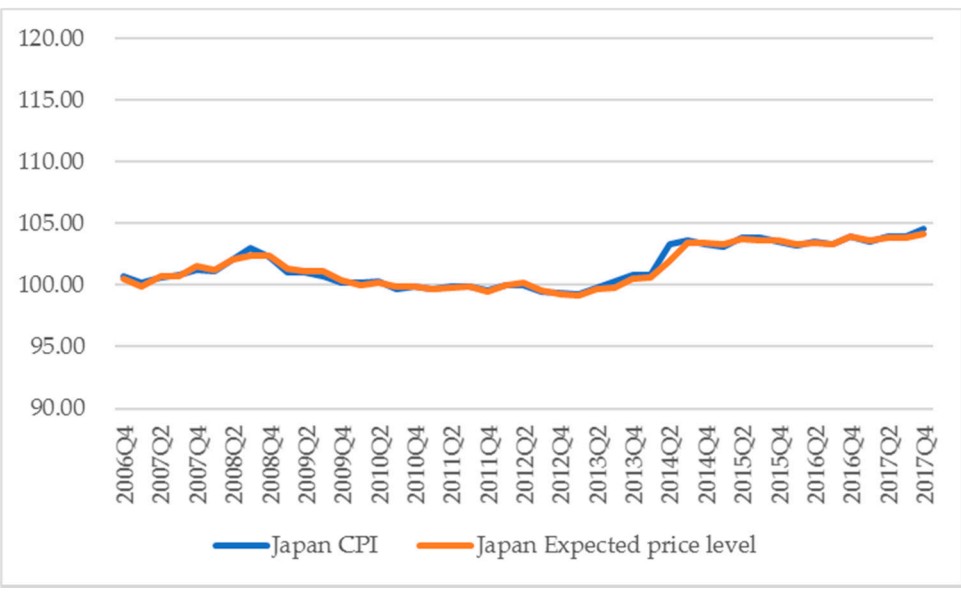

(**c**)

**Figure 2.** *Cont.*

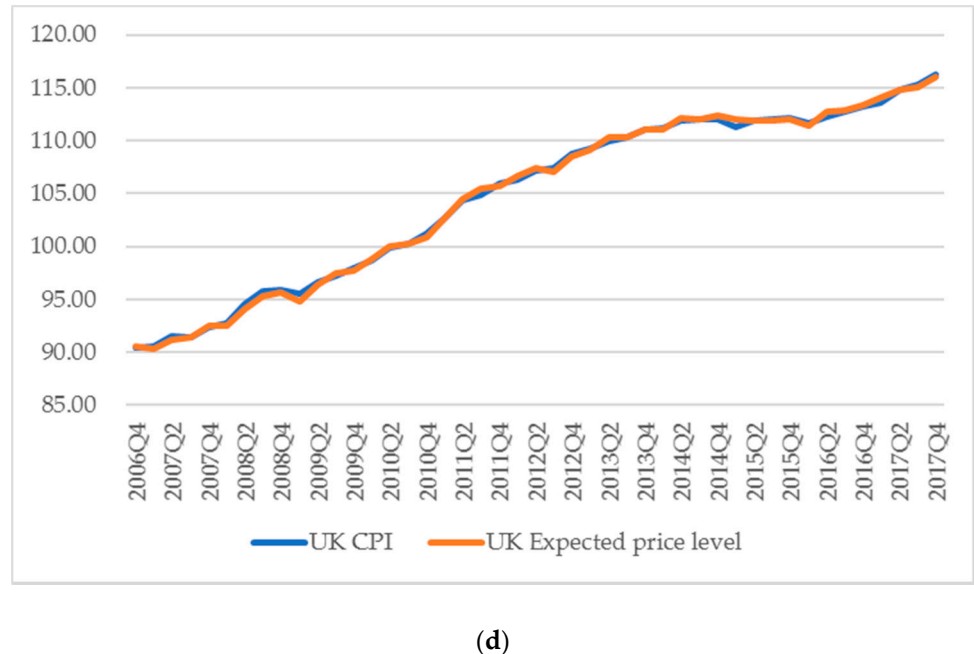

(**d**)

**Figure 2.** (**a**) CPI and expected price level in the United States. (**b**) CPI and expected price level in the euro zone. Notes: CPI is the weighted average of the original euro area. Weights are GDP share. (**c**) CPI and expected price level in Japan. (**d**) CPI and expected price level in the United Kingdom.

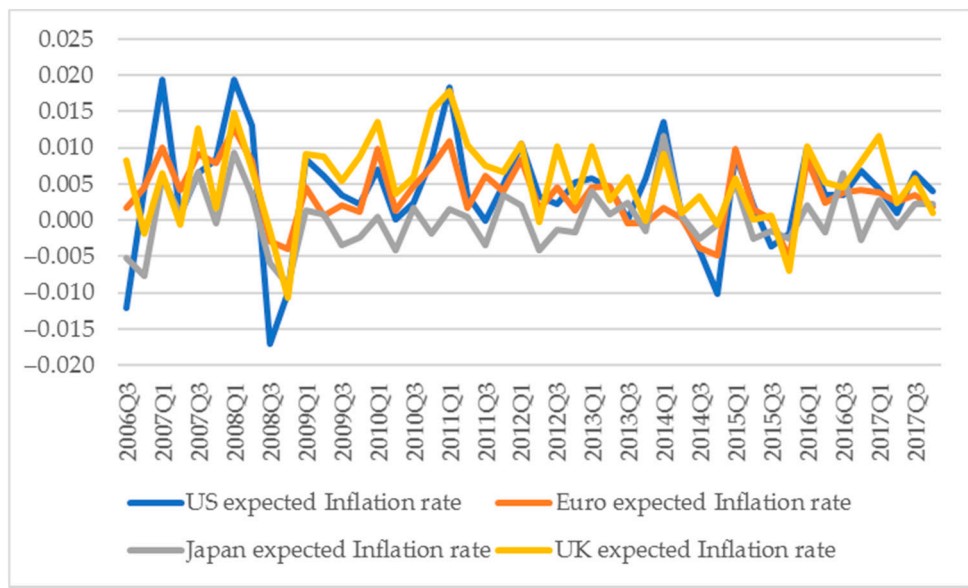

**Figure 3.** Expected inflation rate. Notes: The expected inflation rates are calculated rate of change between actual price level and expected price level estimated under the assumption that the price level of each period follows ARIMA (p, d, q) process. The price level data is CPI.

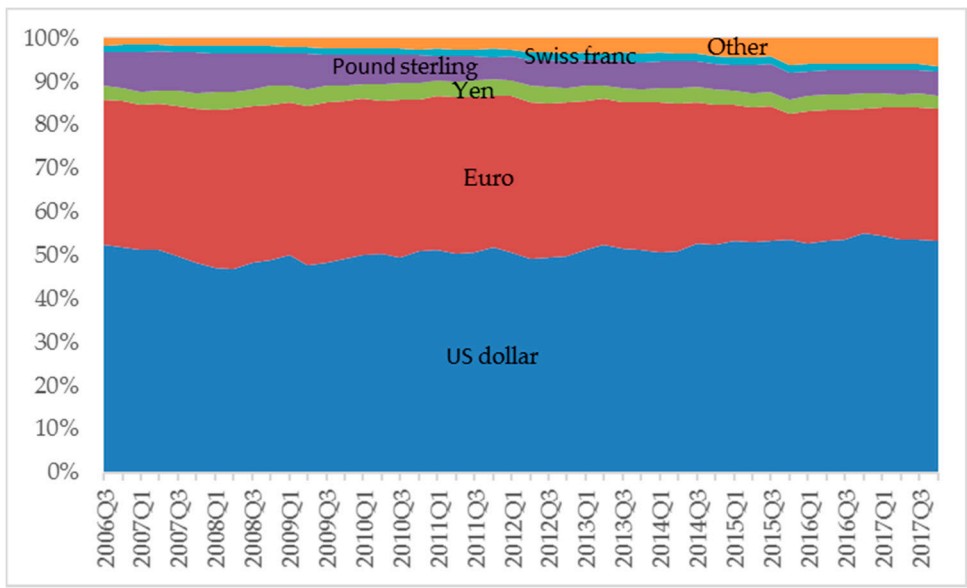

**Figure 4.** Total of domestic currency denominated debt and foreign currency denominated debt of the euro currency market. Data: BIS.

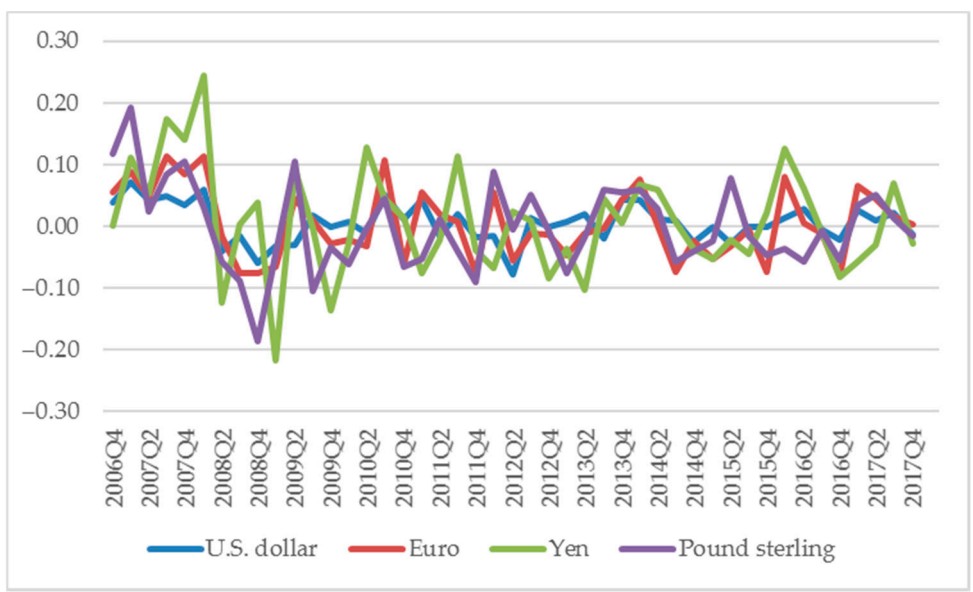

**Figure 5.** Rates of change of share of holdings of the international currencies. Data: BIS, rates of change of total of domestic currency denominated debt and foreign currency denominated debt of the euro currency market.

## 4. Empirical Model

### 4.1. Determinants of Utility of an International Currency

We explain economic variables that can affect utility of international currencies. Firstly, utility of an international currency in the previous period can affect that in the current period if an international currency has inertia in keeping its position. Utility of an international currency may be affected in the same direction as utility of an international currency in the previous period if they have inertia in terms of changes. For example, if utility of an international currency declines, we assumed that the currency is less likely to be used than in the previous period through decline of a medium of exchange

function and economies of scale. In other words, utility of an international currency has inertia in terms of keeping changes in the same direction.

Secondly, supply of liquidity in terms of an international currency can affect its utility. A liquidity risk premium in terms of an international currency is an indicator of a liquidity condition in terms of the relevant international currency or its liquidity shortage. A liquidity shortage reduces utility of an international currency through deteriorating its function as a medium of exchange.

Thirdly, an international currency is more likely to be used in proportion to economic activity in the relevant country. A larger volume of international economic transactions with the relevant country make the international currency more useful in terms of its function as a medium of exchange because of its network externalities. The economic activity in the relevant country and the volume of international economic transactions with the relevant country can be represented by GDP, nominal economic growth rate, real economic growth rate, capitalization, total international trade, total exports, international capital flows, and money stock.

Fourthly, economic agents are likely to prefer a more stable value of currency in holding it as an international currency. Since standard deviation of nominal effective exchange rate is regarded as an indicator of the stability of relevant international currency, it can be a determinant of utility of the relevant international currency. In addition, economic agents are likely to prefer a higher value of currency in holding it as an international currency. An effective exchange rate of an international currency, that is an indicator of a currency value against the other currencies, could be a determinant of utility of the relevant international currency.

### 4.2. A Dynamic Panel Model

We analyze determinants of utility of international currencies by using panel data. In addition, the explanatory variables include a lag term of utility of an international currency as we explained above. For the reasons, we use a dynamic panel data model to analyze determinants of utility of international currencies. Given the above candidates for determinants of an international currency, a dynamic panel model is shown as follows:

$$
\begin{aligned}
&Utility\ of\ international\ currency_{it} \\
&= \hat{b}_1 Utility\ of\ international\ currency_{it-1} + \hat{b}_2 Liquidity\ risk\ premium_{it} \\
&+ \hat{b}_3 Money\ stock\ share_{it} + \hat{b}_4 Relative\ nominal\ economic\ growth_{it} \\
&+ \hat{b}_5 Relative\ real\ economic\ growth_{it} + \hat{b}_6 GDP\ share_{it} \\
&+ \hat{b}_7 Capitalization\ share_{it} + \hat{b}_8 Total\ trade\ share_{it} + \hat{b}_9 Total\ export\ share_{it} \\
&+ \hat{b}_{10} Capital\ flow\ share_{it} + \hat{b}_{11} SD\ of\ Nominal\ effective\ exchange\ rate_{it} \\
&+ \hat{b}_{12} LN\ nominal\ effective\ exchange\ rate_{it} \\
&+ \hat{b}_{13} LN\ real\ effective\ exchange\ rate_{it} + v_i + \varepsilon_{it}
\end{aligned}
\tag{2}
$$

where $v_i$: fixed effects, $\varepsilon_{it}$: disturbance term. We take a first difference of the above model (Equation (2)) and remove fixed effects. Thus, its first difference model is rewritten as follows:

$$
\begin{aligned}
&\Delta Utility\ of\ international\ currency_{it} \\
&= \hat{b}_1 \Delta Utility\ of\ international\ currency_{it-1} + \hat{b}_2 \Delta Liquidity\ risk\ premium_{it} \\
&+ \hat{b}_3 \Delta Money\ stock\ share_{it} + \hat{b}_4 \Delta Relative\ nominal\ economic\ growth_{it} \\
&+ \hat{b}_5 \Delta Relative\ real\ economic\ growth_{it} + \hat{b}_6 \Delta GDP\ share_{it} \\
&+ \hat{b}_7 \Delta Capitalization\ share_{it} + \hat{b}_8 \Delta Total\ trade\ share_{it} + \hat{b}_9 \Delta Total\ export\ share_{it} \\
&+ \hat{b}_{10} \Delta Capital\ flow\ share_{it} + \hat{b}_{11} \Delta SD\ of\ Nominal\ effective\ exchange\ rate_{it} \\
&+ \hat{b}_{12} \Delta LN\ nominal\ effective\ exchange\ rate_{it} \\
&+ \hat{b}_{13} \Delta LN\ real\ effective\ exchange\ rate_{it} + \Delta \varepsilon_{it}
\end{aligned}
\tag{3}
$$

where $\Delta$: difference operator.

There is a correlation between $\Delta Utility\ of\ international\ currency_{it-1}$ and $\Delta \varepsilon_{it}$. Therefore, according to Arellano and Bond (1991), the first difference model is estimated by GMM.

## 5. Sample Period and Data

### 5.1. Sample Period

It is the US dollar, the euro, the Japanese yen, and the British pound that we analyze as international currencies in this paper. The Swiss franc has characteristics that is different from other international currencies and was excluded from the analysis in this paper.

A whole sample period covers a period from 2006Q3 to 2017Q4. The initial period of analysis (2006Q3) is due to constraint of data on the Japanese yen liquidity risk premium. Specifically, we investigate an effect of liquidity shortage on utility of international currencies. In this sample period, the world economy faced US dollar liquidity shortage. Moreover, the Federal Reserve Board (FRB) conducted quantitative easing monetary policy to solve the US dollar liquidity shortage from the end of 2008. The US dollar liquidity shortage could affect utility of the US dollar.

### 5.2. Data for Determinants of Utility of International Currencies

Figure 6a–d show movements in three spreads of London Interbank Offered Rate (LIBOR) (3 months) minus Treasury Bills (TB) rate (3 months), LIBOR (3 months) minus Overnight Indexed Swap (OIS) rate (3 months), and OIS rate (3 months) minus TB rate (3 months). The spread of LIBOR minus OIS rate is regarded as credit risk premium because LIBOR is the interest rate at which banks borrow unsecured funds from other banks. OIS rate is the interest rate at which banks borrow secured funds from other banks. Given that banks mainly face and liquidity risk as well as credit risk, the spread of OIS rate minus TB rate is regarded as liquidity risk premium. The data were obtained from *Datastream.*

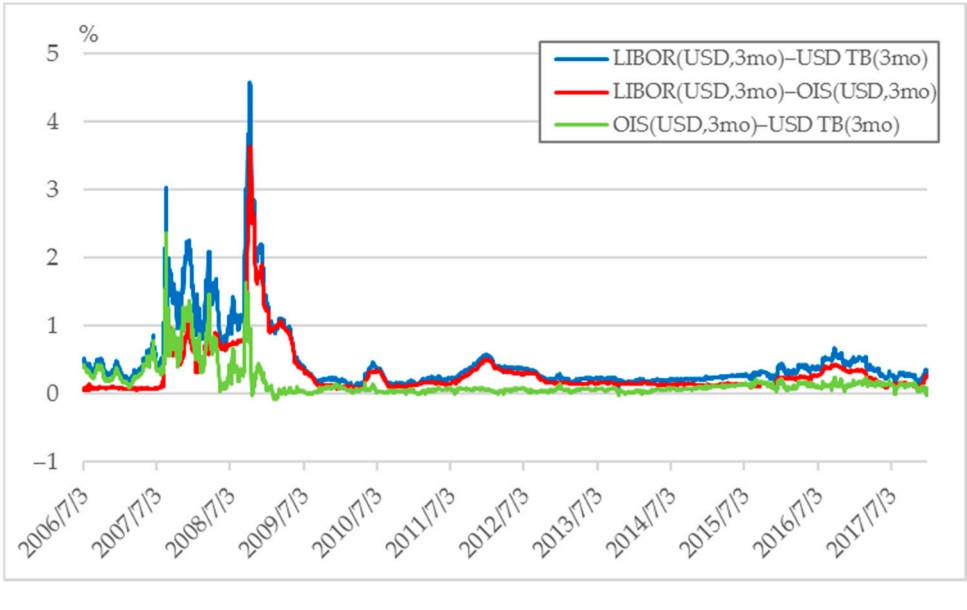

(**a**)

**Figure 6.** *Cont.*

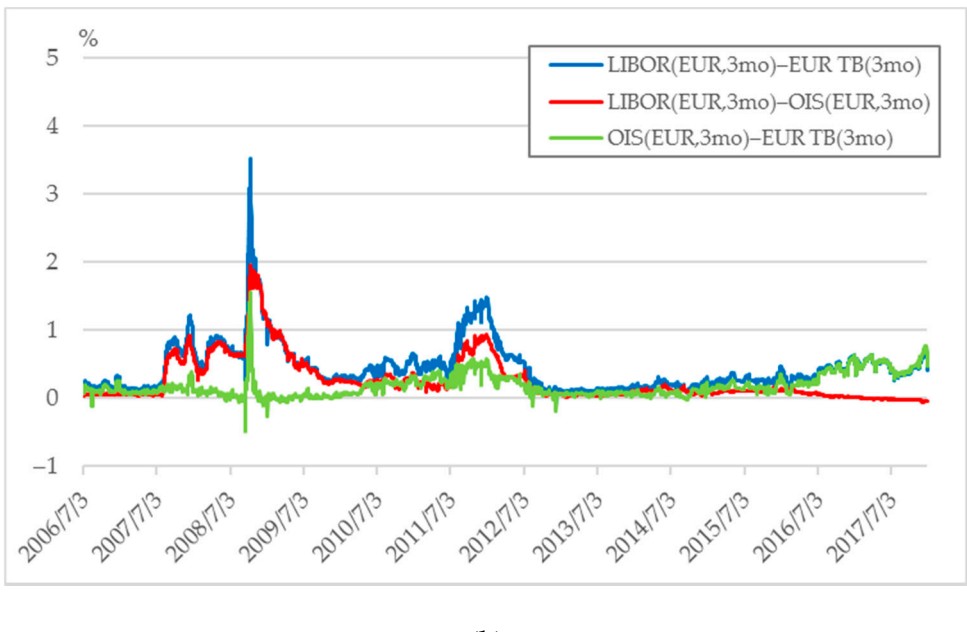

(**b**)

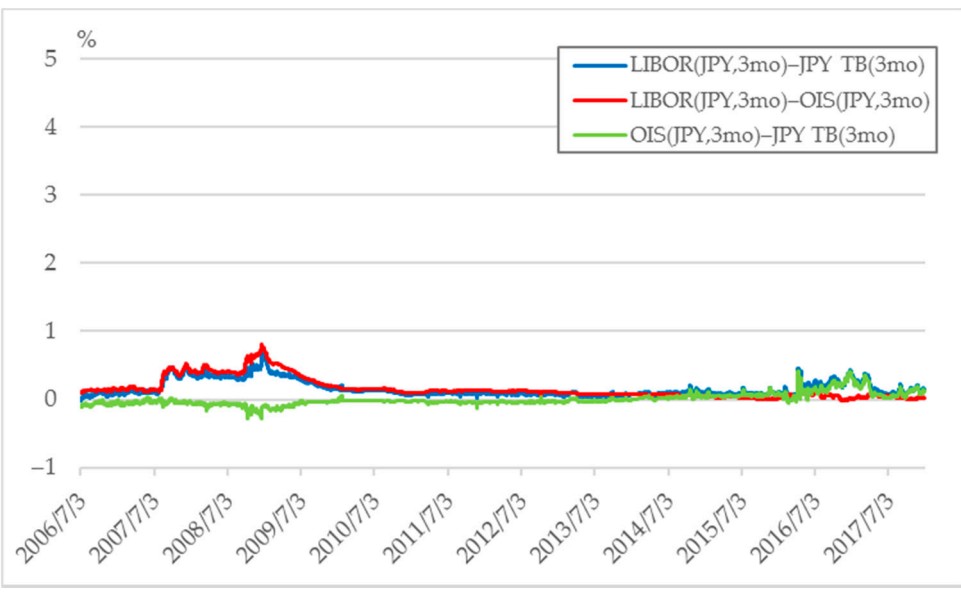

(**c**)

**Figure 6.** *Cont.*

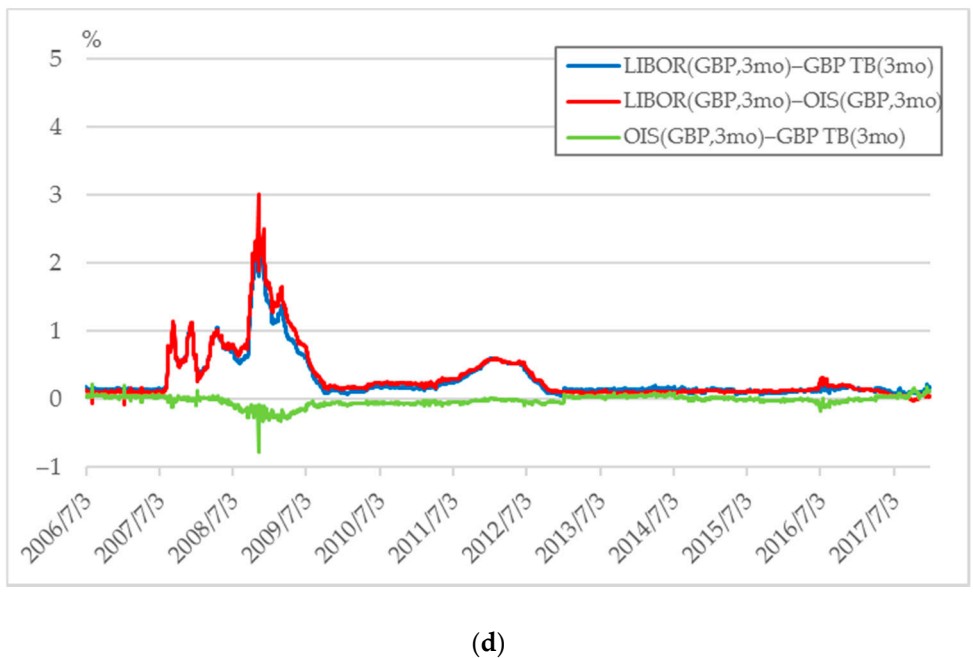

(**d**)

**Figure 6.** (**a**) Credit Risk Premium and Liquidity Risk Premium for the USD. Data: Datastream, Credit risk = London Interbank Offered Rate (LIBOR) (USD, 3 months) minus Overnight Indexed Swap (OIS) rate (USD, 3 months), liquidity risk = OIS minus US Treasury Bills (TB) rate (USD, 3 months). (**b**) Credit Risk Premium and Liquidity Risk Premium for the EUR. Data: Datastream, Credit risk = London Interbank Offered Rate (LIBOR) (EUR, 3 months) minus Overnight Indexed Swap (OIS) rate (EUR, 3 months), liquidity risk = OIS minus yields on German treasury discount paper (Bubills) (EUR TB rate) (euro, 3 months). (**c**) Credit Risk Premium and Liquidity Risk Premium for the JPY. Data: atastream, Credit risk = London Interbank Offered Rate (LIBOR) (JPY, 3 months) minus Overnight Indexed Swap (OIS) rate (JPY, 3 months), liquidity risk = OIS minus yields on Japanese Treasury Discount Bills (JPY TB rate) (JPY, 3 months). (**d**) Credit Risk Premium and Liquidity Risk Premium for the GBP. Data: Datastream, Credit risk = London Interbank Offered Rate (LIBOR) (GBP, 3 months) minus Overnight Indexed Swap (OIS) rate (GBP, 3 months), liquidity risk = OIS minus Yields on UK Government bonds (gilts) (GBP TB rate) (GBP, 3 months).

From Figure 6a, we can find that the US dollar liquidity shortage continues from 2006 to 2008. However, it has decreased to a level smaller than 0.1% since the FRB started quantitative easing monetary policy in late 2008 when it at the same time concluded and extended currency swap arrangements[4] with other major central banks to provide US dollar liquidity to other countries. From Figure 6b, we can find that the euro liquidity shortage from 2006 to 2008 has not occurred except for the Lehman Brothers bankruptcy in September 2008. However, the liquidity risk premium in terms of the euro increased from June 2010 to June 2012. Figure 6c,d do not show any significant increases in liquidity risk premium in terms of the Japanese yen and the British pound during the analysis period. The stable movements in the liquidity risk premium in terms of these currencies are different those in terms of the US dollar and the euro.

Money stock share is a share of money stock of each of the three countries and the region in terms of a total money stock of the three countries and the region. We used seasonally adjusted nominal money stock (M1). The data were obtained from the FRED website.

---

[4]  The FRB concluded new currency swap arrangements with the ECB and the Swiss National Bank on 12 December 2007. Afterwards, it increased amount of currency swap arrangements and concluded them with other central banks.

GDP share is a share of GDP of each of the three countries and the region in terms of a total GDP of the three countries and the region (the United State, the euro Area, Japan, the United Kingdom). We used seasonally adjusted nominal GDP for the calculation. The data were obtained from the *IFS* of IMF website.

Relative nominal economic growth rate and relative real economic growth rate are ratio of GDP growth rate of each of the three countries and the region in terms of an arithmetic average of GDP growth rate of the three countries and the region. Nominal economic growth rate compared to previous quarter was calculated from seasonally adjusted nominal GDP. In addition, we used seasonally adjusted real economic growth rate compared to previous quarter to calculate a relative real economic growth rate. The data were obtained from the Organization for Economic Co-operation and Development (OECD) website.

Capitalization share is a share of capitalization of each of the three countries and the region in terms of a total capitalization of the three countries and the region. We could not obtain the data of United Kingdom in 2010 and quarterly data of the three countries and the region. For the reason, we estimated quarterly data using linear interpolation from annual data. We used data on market capitalization of listed domestic companies for the calculation. The data were obtained from website of the World Bank and the ECB.

Total trade share is a share of trade amount of each of the three countries and the region in terms of a total trade amount of the three countries and the region with the rest of the world. When we sum up the total trade amount for the three countries and the region, we exclude exports and imports among them. Also, total export share is a share of export value of each of the three countries and the region in terms of a total export value of the three countries and the region with the rest of the world. The data were obtained from the *Direction of Trade Statistics* of IMF website.

Capital flow share is a share of international capital flows of each of the three countries and the region in terms of a total international capital flows of the three countries and the region. We could not obtain quarterly data of Japan for 2006Q3 to 2010Q1. For the reason, we estimated quarterly data using linear interpolation from annual data. In this paper, the international capital flows are defined as sum of values of direct investments, portfolio investments, and other investments of net acquisition of financial assets and direct investments, portfolio investments, and other investments of net incurrence of liabilities. The data were obtained from the *Balance of Payments and International Investment Position* of IMF website.

Data on both nominal and real effective exchange rates of each of the currencies are taken logarithms. The data were obtained from BIS website and the *IFS* of IMF website. Table 1 shows mean and standard deviation of difference of each variable.

**Table 1.** Descriptive Statistics of difference of each variable.

| | 4 Countries | | United States | | Euro Area | | Japan | | United Kingdom | |
|---|---|---|---|---|---|---|---|---|---|---|
| | **Mean** | **SD** | **Mean** | **SD** | **Mean** | **SD** | **Mean** | **SD** | **Mean** | **SD** |
| $\Delta$Utility of Currency (1.5%)$_t$ | −0.0001 | 0.1153 | 0.0010 | 0.1666 | −0.0015 | 0.1375 | 0.0004 | 0.0280 | −0.0001 | −0.0001 |
| $\Delta$Utility of Currency (2.0%)$_t$ | 0.0000 | 0.0614 | 0.0010 | 0.0921 | −0.0013 | 0.0743 | 0.0003 | 0.0127 | −0.0002 | −0.0002 |
| $\Delta$Utility of Currency (2.5%)$_t$ | 0.0000 | 0.0441 | 0.0010 | 0.0665 | −0.0012 | 0.0537 | 0.0003 | 0.0092 | −0.0003 | −0.0003 |
| $\Delta$Utility of Currency (3.0%)$_t$ | 0.0000 | 0.0352 | 0.0010 | 0.0530 | −0.0011 | 0.0430 | 0.0002 | 0.0076 | −0.0003 | −0.0003 |
| $\Delta$Utility of Currency (1.5%)$_{t-1}$ | −0.0005 | 0.1221 | 0.0093 | 0.1778 | −0.0068 | 0.1437 | −0.0004 | 0.0285 | −0.0043 | −0.0043 |
| $\Delta$Utility of Currency (2.0%)$_{t-1}$ | −0.0003 | 0.0666 | 0.0063 | 0.1007 | −0.0044 | 0.0787 | −0.0002 | 0.0132 | −0.0028 | −0.0028 |
| $\Delta$Utility of Currency (2.5%)$_{t-1}$ | −0.0002 | 0.0481 | 0.0049 | 0.0730 | −0.0033 | 0.0569 | −0.0002 | 0.0097 | −0.0021 | −0.0021 |
| $\Delta$Utility of Currency (3.0%)$_{t-1}$ | −0.0001 | 0.0383 | 0.0041 | 0.0582 | −0.0027 | 0.0454 | −0.0001 | 0.0080 | −0.0018 | −0.0018 |
| $\Delta$Liquidity Risk Premium$_t$ | 0.0012 | 0.0754 | −0.0036 | 0.1111 | 0.0054 | 0.0873 | 0.0028 | 0.0429 | 0.0004 | 0.0003 |
| $\Delta$Money Stock Share$_t$ | 0.0000 | 0.0087 | 0.0012 | 0.0040 | 0.0003 | 0.0108 | −0.0009 | 0.0122 | −0.0009 | −0.0007 |
| $\Delta$Relative Nominal Economic Growth$_t$ | 0.0000 | 3.9457 | −0.0054 | 1.4781 | 0.0199 | 3.4264 | 0.0064 | 5.0524 | −0.0370 | −0.0208 |
| $\Delta$Relative Real Economic Growth$_t$ | 0.0000 | 6.6128 | 0.0058 | 3.2296 | −0.0024 | 5.4797 | −0.0083 | 10.7073 | 0.0055 | 0.0049 |
| $\Delta$GDP Share$_t$ | 0.0000 | 0.0059 | 0.0012 | 0.0074 | −0.0004 | 0.0049 | −0.0007 | 0.0077 | −0.0002 | −0.0002 |
| $\Delta$Capitalization Share$_t$ | 0.0000 | 0.0042 | 0.0025 | 0.0044 | −0.0017 | 0.0046 | −0.0001 | 0.0039 | −0.0007 | −0.0007 |
| $\Delta$Total Trade Share$_t$ | 0.0000 | 0.0068 | 0.0005 | 0.0072 | −0.0002 | 0.0102 | −0.0002 | 0.0050 | −0.0001 | −0.0001 |
| $\Delta$Total Export Share$_t$ | 0.0000 | 0.0060 | 0.0007 | 0.0044 | −0.0002 | 0.0088 | −0.0004 | 0.0061 | −0.0002 | −0.0002 |
| $\Delta$Capital Flow Share$_t$ | 0.0000 | 0.0049 | 0.0010 | 0.0060 | 0.0006 | 0.0060 | 0.0004 | 0.0028 | −0.0020 | −0.0020 |
| $\Delta$SD of Nominal Effective Exchange Rate$_t$ | 0.0082 | 0.9941 | 0.0107 | 0.6415 | 0.0016 | 0.6647 | 0.0046 | 1.3446 | 0.0089 | 0.0159 |
| $\Delta$LN Nominal Effective Exchange Rate$_t$ | −0.0005 | 0.0340 | 0.0023 | 0.0281 | 0.0007 | 0.0227 | 0.0020 | 0.0494 | −0.0075 | −0.0071 |
| $\Delta$LN Real Effective Exchange Rate$_t$ | −0.0023 | 0.0339 | 0.0015 | 0.0272 | −0.0017 | 0.0227 | −0.0029 | 0.0491 | −0.0065 | −0.0060 |
| number of observations | 172 | | 43 | | 43 | | 43 | | 43 | |

## 6. Empirical Analysis on Determinants of Utility of International Currencies

### 6.1. Expected Effect of Determinant on Utility of International Currencies

We expect that each of determinants affects utility of international currencies in as a direction as follows.

If there is inertia of an international currency in terms of changes in utility, a change in utility of an international currency in the previous period has a positive effect on a change in utility of the international currency in the current period. For example, if utility of an international currency declines, demand for the relevant international currency as one with a function as medium of exchange should decrease. The decrease in the demand for the international currency, in turn, makes utility of an international currency decline further. We investigate a hypothesis that a change in utility of an international currency in the previous period has a positive effect on change in utility of the international currency in the current period.

An increase in the liquidity risk premium in terms of an international currency means liquidity shortage in terms of the relevant international currency. The liquidity shortage reduces convenience of the international currency for economic agents to use for medium of exchange. Thus, an occurrence of liquidity shortage decreases utility of the international currency. The liquidity risk premium, that is indicated by OIS rate minus TB rate, increases when the liquidity shortage worsens. Therefore, an increase in the liquidity risk premium reduces utility of the international currency. We investigate a hypothesis that liquidity risk premium has a negative effect on utility of the international currency. Figures 1a–d and 2a show that both we had both the increase in the US dollar liquidity risk premium in 2008 and the decrease in utility of the US dollar in 2008Q3 simultaneously.

Money stock share, relative nominal economic growth rate, relative real economic growth rate, GDP share, capitalization share, total trade share, total export share, and capital flow share are economic variables that represent relative economic size of the relevant country. Network externalities works in selecting an international currency as medium of exchange. For the reason, a change in an economic size of the relevant country has a positive effect on utility of the international currency. We investigate a hypothesis that coefficients of explanatory variables that represent relative economic size are positive.

As we have already explained, economic agents are likely to prefer a stabler and higher value of currency in holding it as an international currency. The standard deviation and value of effective exchange rate of an international currency, that is an indicator of a currency stable and value against the other currencies, could be a determinant of utility of the relevant international currency. We investigate a hypothesis that a stabler and higher value of an international currency increase. We analyze whether an increase in an effective exchange rate increases utility of the relevant international currency.

### 6.2. Empirical Results

Tables 2–5 show results of the dynamic panel analysis. A head line in Tables represents empirical analysis number. Table 2 shows determinants of utility of international currency supposing that a real interest rate is 1.5%. Coefficients on change in utility of international currencies in the previous period are significantly positive in 26 cases of total 36 cases. The coefficients are estimated from 0.15 to 0.26. Coefficients on change in liquidity risk premium are significantly negative in all of the cases at 1% of significance level. The coefficients are estimated from −0.25 to −0.20. In the analyses 1 to 4 and 12, the coefficients on change in money stock share are positive at 10% of significance level. The coefficients are estimated from 3.50 to 3.72. In the analyses 1 to 12 and 36, the coefficients on change in capital flow share are significantly positive. The coefficients are estimated from 1.21 to 2.93. However, economic scale variables excluding money stock share and capital flow share and effective exchange rate related variables do not satisfy a sign condition or significance levels.

Table 3 shows determinants of utility of international currency supposing that a real interest rate is 2.0%. Coefficients on change in utility of international currencies in the previous period are

significantly positive in 20 cases of total 36 cases. The coefficients are estimated from 0.22 to 0.40. Coefficients on change in liquidity risk premium are significantly negative in all of the cases at 1% of significance level. The coefficients are estimated from $-0.11$ to $-0.08$. In the analysis 49, the coefficients on change in money stock share are positive at 5% of significance level. The coefficient is estimated 0.58. Coefficients on change in capital flow share are significantly positive in all of the cases at the significance level 1%. The coefficients are estimated from 0.86 to 1.71. In the analysis 62, Coefficients on change in nominal effective exchange rate are significantly positive at the significance level 10%. The coefficients are estimated 0.30. However, most of the coefficients on change in economic variables associated with relative economic scale excluding capital flow share and effective exchange rate do not satisfy a sign condition or significance levels.

Table 4 shows determinants of utility of international currency supposing that a real interest rate is 2.5%. Coefficients on change in utility of international currencies in the previous period are significantly positive in 22 cases of total 36 cases. The coefficients are estimated from 0.16 to 0.45. Coefficients on change in liquidity risk premium are significantly negative in all of the cases. The coefficients are estimated from $-0.07$ to $-0.04$. In the analysis 85, the coefficients on change in money stock share are positive at 5% of significance level. The coefficient is estimated 0.42. Coefficients on change in capital flow share are significantly positive in all of the cases except analysis 84. The coefficients are estimated from 0.65 to 1.07. In the analysis 98, the coefficients on change in nominal effective exchange rate are positive at 10% of significance level. The coefficient is estimated 0.24. However, most of the coefficients on change in economic variables associated with relative economic scale excluding capital flow share and effective exchange rate do not satisfy a sign condition or significance levels.

Table 5 shows determinants of utility of international currency supposing that a real interest rate is 3.0%. Coefficients on change in utility of international currencies in the previous period are significantly positive in 23 cases of total 36 cases. The coefficients are estimated to be from 0.06 to 0.43. Coefficients on change in utility of international currencies in the previous period are significantly negative in 14 cases out of 36 cases. The coefficients are estimated to be $-0.04$ and $-0.03$. In the analyses 136 and 144, coefficients on change in capital flow share are significantly positive. The coefficients are estimated to be 0.45 and 0.51. However, most of the coefficients on change in economic variables associated with relative economic scale and effective exchange rate do not satisfy the sign condition or the significance level.

We summarize the above empirical results. Firstly, the coefficients on utility of international currency in the previous period are significantly positive in the many cases. These results suggest that change in utility of an international currency in the previous period in the same direction has effect on change in utility of the international currency in the current period. There is inertia in terms of change in the international monetary system.

Secondly, the coefficients on liquidity risk premium are significantly negative in all of the cases except the real interest rate 3.0%. Even if the real interest rate is 3.0%, liquidity risk premium is significantly negative in about half of cases. The empirical result is consistent with the hypothesis that liquidity risk premium has a negative effect on utility of the international currency. We find that utility of an international currency is affected by liquidity condition or liquidity shortage. Specifically, the liquidity shortage reduces utility of the international currency through a reduction in convenience for economic agents to use the relevant international currency as a medium of exchange.

Thirdly, the coefficients on capital flow share are significant in many cases except in cases of real interest rate 3.0%. Capital flow share represent relative economic scale of the relevant country. Therefore, the above results suggest that utility of the international currency might be affected by changes in economic scale. However, since other economic scale variables did not become significant, the relative change in capital flows may be affecting utility of the international currency.

**Table 2.** Determinants utility of international currency (real interest rate 1.5%).

| | 1 | 2 | 3 | 4 | 5 | 6 | 7 | 8 | 9 | 10 | 11 | 12 | 13 | 14 | 15 | 16 | 17 | 18 |
|---|---|---|---|---|---|---|---|---|---|---|---|---|---|---|---|---|---|---|
| ΔUtility of international currency$_{it-1}$ | 0.14 (0.30) | 0.23 (0.10) | 0.13 (0.28) | 0.23 * (0.09) | 0.16 (0.25) | 0.26 * (0.07) | 0.16 (0.22) | 0.26 * (0.05) | 0.15 (0.28) | 0.24 * (0.08) | 0.14 (0.27) | 0.24 * (0.07) | 0.21 ** (0.01) | 0.20 ** (0.05) | 0.20 ** (0.04) | 0.21 ** (0.02) | 0.21 ** (0.02) | 0.06 (0.43) |
| ΔLiquidity risk premium$_{it}$ | −0.21 *** (0.00) | −0.24 *** (0.00) | −0.21 *** (0.00) | −0.23 *** (0.00) | −0.22 *** (0.00) | −0.25 *** (0.00) | −0.22 *** (0.00) | −0.24 *** (0.00) | −0.22 *** (0.00) | −0.25 *** (0.00) | −0.22 *** (0.00) | −0.24 *** (0.00) | −0.19 *** (0.00) | −0.21 *** (0.00) | −0.21 *** (0.00) | −0.22 *** (0.00) | −0.21 *** (0.00) | −0.23 *** (0.00) |
| ΔMoney stock share$_{it}$ | 3.50 * (0.07) | 3.55 * (0.06) | 3.57 * (0.07) | 3.64 * (0.06) | 3.36 (0.15) | 3.49 (0.13) | 3.44 (0.14) | 3.58 (0.12) | 3.50 (0.11) | 3.62 (0.10) | 3.60 (0.10) | 3.72 * (0.09) | 0.85 (0.19) | | | | | |
| ΔRelative nominal economic growth$_{it}$ | −0.0007 (0.63) | −0.0015 (0.35) | | | −0.0008 (0.59) | −0.0017 (0.33) | | | −0.0009 (0.58) | −0.0017 (0.33) | | | | −0.0020 (0.32) | | | | |
| ΔRelative real economic growth$_{it}$ | | | −0.0005 (0.46) | −0.0006 (0.30) | | | −0.0005 (0.44) | −0.0006 (0.25) | | | −0.0006 (0.41) | −0.0006 (0.23) | | | −0.0003 (0.57) | | | |
| ΔGDP share$_{it}$ | −2.97 (0.32) | −3.71 (0.17) | −2.97 (0.31) | −3.66 (0.15) | −3.69 (0.24) | −4.47 (0.11) | −3.62 (0.23) | −4.38 * (0.10) | −3.46 (0.26) | −4.23 (0.12) | −3.39 (0.26) | −4.14 (0.11) | | | | −0.31 (0.67) | | |
| ΔCapitalization share$_{it}$ | −0.02 (0.98) | −0.29 (0.84) | −0.04 (0.96) | −0.28 (0.84) | 0.13 (0.90) | −0.15 (0.93) | 0.07 (0.94) | −0.18 (0.91) | −0.03 (0.98) | −0.30 (0.85) | −0.10 (0.90) | −0.35 (0.82) | | | | | −0.03 (0.88) | |
| ΔTotal trade share$_{it}$ | −5.78 *** (0.00) | | −5.83 *** (0.00) | | −5.71 *** (0.00) | | −5.74 *** (0.00) | | −5.72 *** (0.00) | | −5.75 *** (0.00) | | | | | | | −4.67 *** (0.00) |
| ΔTotal export share$_{it}$ | | −3.45 * (0.07) | | −3.56 * (0.08) | | −3.41 * (0.08) | | −3.50 * (0.08) | | −3.48 * (0.08) | | −3.56 * (0.08) | | | | | | |
| ΔCapital flow share$_{it}$ | 2.36 *** (0.00) | 2.36 ** (0.02) | 2.34 *** (0.00) | 2.30 ** (0.03) | 2.85 *** (0.00) | 2.93 *** (0.01) | 2.78 *** (0.00) | 2.83 ** (0.02) | 2.83 *** (0.00) | 2.91 ** (0.02) | 2.76 *** (0.00) | 2.81 ** (0.04) | | | | | | |
| ΔSD of Nominal effective exchange rate$_{it}$ | −0.01 (0.16) | −0.01 ** (0.04) | −0.01 (0.16) | −0.01 ** (0.04) | | | | | | | | | −0.02 (0.25) | −0.02 (0.24) | −0.02 (0.25) | −0.02 (0.26) | −0.02 (0.23) | −0.01 (0.36) |
| ΔLN Nominal effective exchange rate$_{it}$ | | | | | 0.15 (0.65) | 0.13 (0.72) | 0.14 (0.65) | 0.13 (0.71) | | | | | | | | | | |
| ΔLN Real effective exchange rate$_{it}$ | | | | | | | | | 0.05 (0.87) | 0.04 (0.92) | 0.05 (0.89) | 0.04 (0.91) | | | | | | |
| Sargan test | 0.95 | 0.97 | 0.95 | 0.97 | 0.99 | 0.99 | 0.99 | 0.99 | 0.98 | 0.99 | 0.98 | 0.99 | 0.97 | 0.98 | 0.98 | 0.97 | 0.98 | 0.87 |
| AR(1) serial correlation test | 0.08 * | 0.10 * | 0.08 * | 0.10 * | 0.08 * | 0.10 | 0.08 * | 0.10 | 0.08 * | 0.10 * | 0.08 * | 0.10 * | 0.09 * | 0.10 * | 0.10 * | 0.10 * | 0.10 | 0.08 * |
| AR(2) serial correlation test | 0.47 | 0.20 | 0.58 | 0.22 | 0.97 | 0.23 | 0.88 | 0.27 | 0.86 | 0.23 | 0.95 | 0.26 | 0.18 | 0.19 | 0.19 | 0.18 | 0.18 | 0.13 |

**Table 2.** *Cont.*

| | 19 | 20 | 21 | 22 | 23 | 24 | 25 | 26 | 27 | 28 | 29 | 30 | 31 | 32 | 33 | 34 | 35 | 36 |
|---|---|---|---|---|---|---|---|---|---|---|---|---|---|---|---|---|---|---|
| ΔUtility of international currency$_{it-1}$ | 0.15 ** (0.05) | 0.22 ** (0.01) | 0.21 ** (0.04) | 0.23 ** (0.02) | 0.23 ** (0.02) | 0.23 ** (0.02) | 0.23 ** (0.04) | 0.07 (0.50) | 0.16 * (0.09) | 0.25 *** (0.01) | 0.20 ** (0.04) | 0.22 ** (0.03) | 0.22 ** (0.03) | 0.23 ** (0.02) | 0.23 ** (0.04) | 0.07 (0.51) | 0.16 * (0.10) | 0.25 ** (0.01) |
| ΔLiquidity risk premium$_{it}$ | −0.25 *** (0.00) | −0.21 *** (0.00) | −0.20 *** (0.00) | −0.21 *** (0.00) | −0.20 *** (0.00) | −0.21 *** (0.00) | −0.20 *** (0.00) | −0.22 *** (0.00) | −0.24 *** (0.00) | −0.21 *** (0.00) | −0.20 *** (0.00) | −0.21 *** (0.00) | −0.21 *** (0.00) | −0.21 *** (0.00) | −0.20 *** (0.00) | −0.22 *** (0.00) | −0.24 *** (0.00) | −0.22 *** (0.00) |
| ΔMoney stock share$_{it}$ | | | 0.61 (0.47) | | | | | | | | 0.90 (0.23) | | | | | | | |
| ΔRelative nominal economic growth$_{it}$ | | | | −0.0019 (0.35) | | | | | | | | −0.0020 (0.34) | | | | | | |
| ΔRelative real economic growth$_{it}$ | | | | | −0.0002 (0.56) | | | | | | | | −0.0002 (0.55) | | | | | |
| ΔGDP share$_{it}$ | | | | | | −1.83 *** (0.00) | | | | | | | | −1.68 ** (0.01) | | | | |
| ΔCapitalization share$_{it}$ | | | | | | | −0.31 (0.57) | | | | | | | | −0.23 (0.71) | | | |
| ΔTotal trade share$_{it}$ | | | | | | | | −5.57 *** (0.00) | | | | | | | | −5.52 *** (0.00) | | |
| ΔTotal export share$_{it}$ | −2.70 (0.11) | | | | | | | | −3.56 * (0.07) | | | | | | | | −3.55 * (0.06) | |
| ΔCapital flow share$_{it}$ | | 1.00 (0.26) | | | | | | | | 1.05 (0.15) | | | | | | | | 1.21 * (0.06) |
| ΔSD of Nominal effective exchange rate$_{it}$ | −0.01 (0.26) | −0.02 (0.16) | | | | | | | | | | | | | | | | |
| ΔLN Nominal effective exchange rate$_{it}$ | | | 0.01 (0.97) | 0.14 (0.58) | 0.17 (0.50) | 0.45 (0.10) | 0.17 (0.51) | 0.43 (0.18) | 0.33 (0.29) | 0.11 (0.67) | | | | | | | | |
| ΔLN Real effective exchange rate$_{it}$ | | | | | | | | | | | −0.09 (0.78) | 0.09 (0.72) | 0.12 (0.62) | 0.39 (0.15) | 0.12 (0.64) | 0.40 (0.19) | 0.30 (0.32) | 0.04 (0.86) |
| Sargan test | 0.94 | 0.98 | 0.98 | 1.00 | 1.00 | 1.00 | 1.00 | 0.98 | 0.99 | 1.00 | 0.97 | 1.00 | 1.00 | 1.00 | 1.00 | 0.98 | 0.99 | 1.00 |
| AR(1) serial correlation test | 0.09 * | 0.09 * | 0.09 * | 0.10 * | 0.10 * | 0.10 | 0.10 * | 0.08 * | 0.09 * | 0.09 * | 0.09 * | 0.10 * | 0.10 * | 0.10 | 0.10 | 0.08 * | 0.09 * | 0.09 * |
| AR(2) serial correlation test | 0.16 | 0.20 | 0.21 | 0.26 | 0.25 | 0.32 | 0.24 | 0.51 | 0.16 | 0.27 | 0.21 | 0.26 | 0.25 | 0.32 | 0.24 | 0.47 | 0.16 | 0.26 |

The parentheses are *p*-value. *, **, *** are significance level 10%, 5%, 1%. Instrument variables for period t are utility of the international currency of periods t-3. The null hypothesis of Sargan test is that over-identification is valid. The null hypothesis of AR(1) and AR(2) serial correlation test is that there is no serial correlation.

**Table 3.** Determinants utility of international currency (real interest rate 2.0%).

| | 37 | 38 | 39 | 40 | 41 | 42 | 43 | 44 | 45 | 46 | 47 | 48 | 49 | 50 | 51 | 52 | 53 | 54 |
|---|---|---|---|---|---|---|---|---|---|---|---|---|---|---|---|---|---|---|
| ΔUtility of international currency$_{it-1}$ | 0.20 (0.38) | 0.30 (0.22) | 0.19 (0.35) | 0.30 (0.20) | 0.25 (0.32) | 0.37 (0.18) | 0.25 (0.28) | 0.37 (0.14) | 0.20 (0.35) | 0.31 (0.18) | 0.19 (0.31) | 0.30 (0.14) | 0.31 *** (0.01) | 0.29 ** (0.03) | 0.28 ** (0.02) | 0.31 ** (0.01) | 0.32 ** (0.01) | 0.12 (0.20) |
| ΔLiquidity risk premium$_{it}$ | −0.08 *** (0.00) | −0.10 *** (0.00) | −0.08 *** (0.00) | −0.10 *** (0.00) | −0.10 *** (0.00) | −0.11 *** (0.00) | −0.10 *** (0.00) | −0.11 *** (0.00) | −0.10 *** (0.00) | −0.11 *** (0.00) | −0.09 *** (0.00) | −0.11 *** (0.00) | −0.07 *** (0.00) | −0.08 *** (0.00) | −0.08 *** (0.00) | −0.08 *** (0.00) | −0.08 *** (0.00) | −0.09 *** (0.00) |
| ΔMoney stock share$_{it}$ | 1.54 (0.17) | 1.59 (0.16) | 1.57 (0.16) | 1.64 (0.13) | 1.40 (0.36) | 1.54 (0.32) | 1.44 (0.34) | 1.59 (0.29) | 1.41 (0.30) | 1.54 (0.27) | 1.45 (0.28) | 1.59 (0.23) | 0.58 ** (0.05) | | | | | |
| ΔRelative nominal economic growth$_{it}$ | −0.0007 (0.46) | −0.0012 (0.27) | | | −0.0008 (0.48) | −0.0014 (0.28) | | | −0.0008 (0.47) | −0.0013 (0.27) | | | | −0.0014 (0.24) | | | | |
| ΔRelative real economic growth$_{it}$ | | | −0.0002 (0.56) | −0.0003 (0.40) | | | −0.0002 (0.54) | −0.0003 (0.34) | | | −0.0002 (0.51) | −0.0003 (0.30) | | | −0.0001 (0.71) | | | |
| ΔGDP share$_{it}$ | −0.90 (0.60) | −1.36 (0.38) | −0.86 (0.59) | −1.29 (0.35) | −1.51 (0.42) | −2.00 (0.25) | −1.46 (0.40) | −1.94 (0.21) | −1.19 (0.46) | −1.66 (0.25) | −1.14 (0.44) | −1.59 (0.22) | | | | 0.20 (0.43) | | |
| ΔCapitalization share$_{it}$ | 0.09 (0.85) | −0.08 (0.93) | 0.10 (0.83) | −0.05 (0.95) | 0.27 (0.67) | 0.09 (0.93) | 0.25 (0.68) | 0.09 (0.93) | 0.12 (0.80) | −0.05 (0.96) | 0.09 (0.83) | −0.06 (0.94) | | | | | 0.32 (0.19) | |
| ΔTotal trade share$_{it}$ | −3.45 *** (0.00) | | −3.51 *** (0.00) | | −3.37 *** (0.01) | | −3.42 *** (0.00) | | −3.46 *** (0.01) | | −3.51 *** (0.00) | | | | | | | −2.61 *** (0.00) |
| ΔTotal export share$_{it}$ | | −2.12 * (0.06) | | −2.23 * (0.05) | | −1.97 * (0.07) | | −2.06 * (0.05) | | −2.13 * (0.06) | | −2.22 * (0.05) | | | | | | |
| ΔCapital flow share$_{it}$ | 1.32 *** (0.00) | 1.35 *** (0.00) | 1.28 *** (0.00) | 1.28 *** (0.00) | 1.63 *** (0.00) | 1.71 *** (0.00) | 1.57 *** (0.00) | 1.62 *** (0.00) | 1.51 *** (0.00) | 1.59 *** (0.00) | 1.46 *** (0.00) | 1.50 *** (0.01) | | | | | | |
| ΔSD of Nominal effective exchange rate$_{it}$ | 0.00 (0.18) | −0.01 ** (0.03) | −0.01 (0.18) | −0.01 ** (0.03) | | | | | | | | | −0.01 (0.24) | −0.01 (0.26) | −0.01 (0.26) | −0.01 (0.25) | −0.01 (0.21) | −0.01 (0.40) |
| ΔLN Nominal effective exchange rate$_{it}$ | | | | | 0.14 (0.49) | 0.12 (0.60) | 0.15 (0.47) | 0.13 (0.57) | | | | | | | | | | |
| ΔLN Real effective exchange rate$_{it}$ | | | | | | | | | 0.08 (0.71) | 0.05 (0.82) | 0.08 (0.69) | 0.06 (0.79) | | | | | | |
| Sargan test | 0.80 | 0.88 | 0.80 | 0.89 | 0.94 | 0.97 | 0.95 | 0.98 | 0.84 | 0.92 | 0.84 | 0.92 | 0.91 | 0.89 | 0.89 | 0.91 | 0.92 | 0.51 |
| AR(1) serial correlation test | 0.09 * | 0.13 | 0.09 * | 0.13 | 0.11 | 0.15 | 0.11 | 0.15 | 0.10 * | 0.13 | 0.09 * | 0.13 | 0.10 | 0.12 | 0.12 | 0.11 | 0.12 | 0.11 |
| AR(2) serial correlation test | 0.55 | 0.26 | 0.62 | 0.27 | 0.99 | 0.23 | 0.91 | 0.23 | 0.70 | 0.22 | 0.77 | 0.22 | 0.22 | 0.21 | 0.20 | 0.21 | 0.21 | 0.16 |

**Table 3.** *Cont.*

| | 55 | 56 | 57 | 58 | 59 | 60 | 61 | 62 | 63 | 64 | 65 | 66 | 67 | 68 | 69 | 70 | 71 | 72 |
|---|---|---|---|---|---|---|---|---|---|---|---|---|---|---|---|---|---|---|
| ΔUtility of international currency$_{it-1}$ | 0.22 ** (0.03) | 0.34 ** (0.01) | 0.31 ** (0.01) | 0.36 *** (0.00) | 0.36 *** (0.00) | 0.37 *** (0.00) | 0.38 ** (0.01) | 0.16 (0.24) | 0.27 * (0.05) | 0.40 *** (0.01) | 0.29 ** (0.01) | 0.35 *** (0.00) | 0.35 *** (0.00) | 0.36 *** (0.00) | 0.37 ** (0.02) | 0.16 (0.25) | 0.26 * (0.06) | 0.40 ** (0.01) |
| ΔLiquidity risk premium$_{it}$ | −0.09 *** (0.00) | −0.09 *** (0.00) | −0.08 *** (0.00) | −0.09 *** (0.00) | −0.09 *** (0.00) | −0.08 *** (0.00) | −0.09 *** (0.00) | −0.09 *** (0.00) | −0.10 *** (0.00) | −0.10 *** (0.00) | −0.08 *** (0.00) | −0.09 *** (0.00) | −0.09 *** (0.00) | −0.09 *** (0.00) | −0.09 *** (0.00) | −0.09 *** (0.00) | −0.10 *** (0.00) | −0.10 *** (0.00) |
| ΔMoney stock share$_{it}$ | | | 0.29 (0.61) | | | | | | | | 0.44 (0.43) | | | | | | | |
| ΔRelative nominal economic growth$_{it}$ | | | | −0.0014 (0.28) | | | | | | | | −0.0014 (0.27) | | | | | | |
| ΔRelative real economic growth$_{it}$ | | | | | −0.0001 (0.75) | | | | | | | | −0.0001 (0.76) | | | | | |
| ΔGDP share$_{it}$ | | | | | | −0.62 *** (0.00) | | | | | | | | −0.49 ** (0.02) | | | | |
| ΔCapitalization share$_{it}$ | | | | | | | 0.15 (0.81) | | | | | | | | 0.21 (0.75) | | | |
| ΔTotal trade share$_{it}$ | | | | | | | | −3.29 *** (0.00) | | | | | | | | −3.25 *** (0.00) | | |
| ΔTotal export share$_{it}$ | −1.45 * (0.07) | | | | | | | | −2.01 * (0.05) | | | | | | | | −2.02 ** (0.05) | |
| ΔCapital flow share$_{it}$ | | 0.89 *** (0.00) | | | | | | | | 0.86 *** (0.00) | | | | | | | | 0.97 *** (0.00) |
| ΔSD of Nominal effective exchange rate$_{it}$ | −0.01 (0.29) | −0.01 * (0.09) | | | | | | | | | | | | | | | | |
| ΔLN Nominal effective exchange rate$_{it}$ | | | 0.07 (0.76) | 0.12 (0.36) | 0.14 (0.28) | 0.25 (0.17) | 0.14 (0.36) | 0.30 * (0.10) | 0.24 (0.16) | 0.09 (0.52) | | | | | | | | |
| ΔLN Real effective exchange rate$_{it}$ | | | | | | | | | | | 0.00 (0.99) | 0.09 (0.47) | 0.11 (0.36) | 0.20 (0.25) | 0.10 (0.48) | 0.29 (0.11) | 0.22 (0.17) | 0.05 (0.72) |
| Sargan test | 0.79 | 0.94 | 0.95 | 0.99 | 0.99 | 0.99 | 0.99 | 0.93 | 0.98 | 0.99 | 0.91 | 0.98 | 0.99 | 0.99 | 0.99 | 0.92 | 0.97 | 0.99 |
| AR(1) serial correlation test | 0.12 | 0.11 | 0.11 | 0.11 | 0.11 | 0.12 | 0.11 | 0.10 * | 0.11 | 0.10 | 0.11 | 0.11 | 0.12 | 0.12 | 0.12 | 0.10 | 0.12 | 0.11 |
| AR(2) serial correlation test | 0.18 | 0.22 | 0.21 | 0.25 | 0.24 | 0.26 | 0.24 | 0.72 | 0.17 | 0.25 | 0.21 | 0.25 | 0.23 | 0.25 | 0.24 | 0.66 | 0.17 | 0.25 |

The parentheses are *p*-value. *, **, *** are significance level 10%, 5%, 1%. Instrument variables for period t are utility of the international currency of periods t-3. The null hypothesis of Sargan test is that over-identification is valid. The null hypothesis of AR(1) and AR(2) serial correlation test is that there is no serial correlation.

**Table 4.** Determinants utility of international currency (real interest rate 2.5%).

| | 73 | 74 | 75 | 76 | 77 | 78 | 79 | 80 | 81 | 82 | 83 | 84 | 85 | 86 | 87 | 88 | 89 | 90 |
|---|---|---|---|---|---|---|---|---|---|---|---|---|---|---|---|---|---|---|
| ΔUtility of international currency$_{it-1}$ | 0.12 (0.55) | 0.23 (0.37) | 0.12 (0.52) | 0.22 (0.33) | 0.20 (0.48) | 0.32 (0.34) | 0.21 (0.42) | 0.33 (0.29) | 0.11 (0.50) | 0.22 (0.29) | 0.11 (0.43) | 0.22 (0.23) | 0.32 *** (0.00) | 0.31 ** (0.03) | 0.30 ** (0.03) | 0.33 ** (0.01) | 0.34 *** (0.01) | 0.16 ** (0.05) |
| ΔLiquidity risk premium$_{it}$ | −0.04 * (0.08) | −0.05 * (0.09) | −0.04 * (0.10) | −0.05 * (0.09) | −0.05 ** (0.03) | −0.07 ** (0.04) | −0.05 ** (0.03) | −0.06 ** (0.04) | −0.05 * (0.06) | −0.06 * (0.06) | −0.05 * (0.07) | −0.06 * (0.06) | −0.03 * (0.06) | −0.04 ** (0.01) | −0.04 ** (0.02) | −0.04 ** (0.02) | −0.04 ** (0.01) | −0.04 *** (0.00) |
| ΔMoney stock share$_{it}$ | 0.79 (0.29) | 0.82 (0.26) | 0.82 (0.26) | 0.86 (0.22) | 0.69 (0.53) | 0.80 (0.47) | 0.73 (0.50) | 0.85 (0.43) | 0.64 (0.46) | 0.75 (0.40) | 0.67 (0.42) | 0.79 (0.35) | 0.42 ** (0.04) | | | | | |
| ΔRelative nominal economic growth$_{it}$ | −0.0005 (0.45) | −0.0009 (0.25) | | | −0.0006 (0.50) | −0.0010 (0.30) | | | −0.0005 (0.46) | −0.0009 (0.24) | | | | −0.0011 (0.22) | | | | |
| ΔRelative real economic growth$_{it}$ | | | −0.0001 (0.76) | −0.0001 (0.62) | | | −0.0001 (0.69) | −0.0001 (0.45) | | | −0.0001 (0.71) | −0.0001 (0.49) | | | −0.0001 (0.76) | | | |
| ΔGDP share$_{it}$ | −0.09 (0.94) | −0.39 (0.68) | −0.07 (0.94) | −0.36 (0.66) | −0.60 (0.65) | −0.93 (0.44) | −0.60 (0.62) | −0.92 (0.40) | −0.27 (0.76) | −0.59 (0.46) | −0.25 (0.76) | −0.56 (0.42) | | | | 0.25 (0.14) | | |
| ΔCapitalization share$_{it}$ | 0.00 (0.99) | −0.14 (0.75) | 0.01 (0.94) | −0.12 (0.79) | 0.16 (0.62) | 0.01 (0.99) | 0.17 (0.62) | 0.02 (0.97) | 0.04 (0.82) | −0.11 (0.81) | 0.03 (0.85) | −0.11 (0.80) | | | | | 0.27 ** (0.02) | |
| ΔTotal trade share$_{it}$ | −2.72 *** (0.00) | | −2.76 *** (0.00) | | −2.63 ** (0.01) | | −2.65 *** (0.01) | | −2.77 *** (0.00) | | −2.79 *** (0.00) | | | | | | | −1.82 *** (0.00) |
| ΔTotal export share$_{it}$ | | −1.83 * (0.05) | | −1.89 ** (0.05) | | −1.64 * (0.07) | | −1.69 * (0.06) | | −1.84 ** (0.05) | | −1.89 ** (0.04) | | | | | | |
| ΔCapital flow share$_{it}$ | 0.80 *** (0.00) | 0.81 ** (0.02) | 0.78 *** (0.01) | 0.76 * (0.06) | 1.02 *** (0.00) | 1.07 *** (0.00) | 0.99 *** (0.00) | 1.02 *** (0.00) | 0.89 ** (0.02) | 0.94 ** (0.04) | 0.87 ** (0.05) | 0.89 (0.10) | | | | | | |
| ΔSD of Nominal effective exchange rate$_{it}$ | 0.00 (0.35) | 0.00 (0.16) | 0.00 (0.34) | 0.00 (0.15) | | | | | | | | | −0.01 (0.24) | −0.01 (0.26) | −0.01 (0.26) | −0.01 (0.23) | −0.01 (0.20) | 0.00 (0.38) |
| ΔLN Nominal effective exchange rate$_{it}$ | | | | | 0.12 (0.39) | 0.09 (0.52) | 0.12 (0.36) | 0.10 (0.48) | | | | | | | | | | |
| ΔLN Real effective exchange rate$_{it}$ | | | | | | | | | 0.07 (0.57) | 0.04 (0.74) | 0.07 (0.54) | 0.05 (0.70) | | | | | | |
| Sargan test | 0.54 | 0.71 | 0.57 | 0.74 | 0.81 | 0.90 | 0.84 | 0.93 | 0.54 | 0.74 | 0.56 | 0.77 | 0.89 | 0.85 | 0.86 | 0.90 | 0.90 | 0.42 |
| AR(1) serial correlation test | 0.09 * | 0.15 | 0.09 * | 0.14 | 0.14 | 0.20 | 0.13 | 0.19 | 0.10 * | 0.14 | 0.10 * | 0.13 | 0.10 * | 0.12 | 0.13 | 0.11 | 0.12 | 0.13 |
| AR(2) serial correlation test | 0.35 | 0.25 | 0.38 | 0.26 | 0.63 | 0.23 | 0.73 | 0.23 | 0.32 | 0.20 | 0.34 | 0.19 | 0.24 | 0.22 | 0.22 | 0.23 | 0.23 | 0.18 |

**Table 4.** *Cont.*

| | 91 | 92 | 93 | 94 | 95 | 96 | 97 | 98 | 99 | 100 | 101 | 102 | 103 | 104 | 105 | 106 | 107 | 108 |
|---|---|---|---|---|---|---|---|---|---|---|---|---|---|---|---|---|---|---|
| ΔUtility of international currency$_{it-1}$ | 0.24 ** (0.02) | 0.36 ** (0.02) | 0.34 ** (0.01) | 0.42 *** (0.00) | 0.42 *** (0.00) | 0.43 *** (0.00) | 0.44 ** (0.02) | 0.21 (0.18) | 0.31 * (0.08) | 0.45 ** (0.03) | 0.31 *** (0.01) | 0.41 *** (0.01) | 0.41 *** (0.00) | 0.42 *** (0.00) | 0.44 ** (0.02) | 0.21 (0.20) | 0.31 * (0.09) | 0.45 ** (0.04) |
| ΔLiquidity risk premium$_{it}$ | −0.05 *** (0.00) | −0.04 ** (0.05) | −0.04 ** (0.03) | −0.05 *** (0.00) | −0.05 *** (0.00) | −0.05 *** (0.01) | −0.05 *** (0.00) | −0.05 *** (0.00) | −0.06 *** (0.01) | −0.06 *** (0.00) | −0.04 ** (0.04) | −0.05 *** (0.00) | −0.05 *** (0.00) | −0.05 *** (0.00) | −0.05 *** (0.00) | −0.05 *** (0.00) | −0.06 *** (0.01) | −0.06 *** (0.00) |
| ΔMoney stock share$_{it}$ | | | 0.17 (0.69) | | | | | | | | 0.27 (0.52) | | | | | | | |
| ΔRelative nominal economic growth$_{it}$ | | | | −0.0011 (0.27) | | | | | | | | −0.0011 (0.26) | | | | | | |
| ΔRelative real economic growth$_{it}$ | | | | | −0.0001 (0.81) | | | | | | | | 0.0000 (0.82) | | | | | |
| ΔGDP share$_{it}$ | | | | | | −0.31 * (0.09) | | | | | | | | −0.20 (0.40) | | | | |
| ΔCapitalization share$_{it}$ | | | | | | | 0.17 (0.74) | | | | | | | | 0.22 (0.70) | | | |
| ΔTotal trade share$_{it}$ | | | | | | | | −2.39 *** (0.00) | | | | | | | | −2.37 *** (0.00) | | |
| ΔTotal export share$_{it}$ | −1.03 * (0.08) | | | | | | | | −1.44 * (0.06) | | | | | | | | −1.46 ** (0.05) | |
| ΔCapital flow share$_{it}$ | | 0.66 *** (0.00) | | | | | | | | 0.65 ** (0.01) | | | | | | | | 0.73 *** (0.01) |
| ΔSD of Nominal effective exchange rate$_{it}$ | 0.00 (0.29) | 0.00 * (0.07) | | | | | | | | | | | | | | | | |
| ΔLN Nominal effective exchange rate$_{it}$ | | | 0.07 (0.69) | 0.10 (0.30) | 0.12 (0.22) | 0.17 (0.21) | 0.11 (0.33) | 0.24 * (0.08) | 0.18 (0.12) | 0.07 (0.48) | | | | | | | | |
| ΔLN Real effective exchange rate$_{it}$ | | | | | | | | | | | 0.02 (0.90) | 0.08 (0.40) | 0.10 (0.30) | 0.13 (0.34) | 0.08 (0.45) | 0.23 * (0.09) | 0.17 (0.14) | 0.04 (0.70) |
| Sargan test | 0.75 | 0.90 | 0.95 | 0.99 | 0.99 | 0.99 | 0.99 | 0.93 | 0.98 | 0.99 | 0.89 | 0.98 | 0.99 | 0.99 | 0.99 | 0.92 | 0.98 | 0.98 |
| AR(1) serial correlation test | 0.13 | 0.11 | 0.11 | 0.11 | 0.11 | 0.12 | 0.11 | 0.11 | 0.13 | 0.11 | 0.10 | 0.11 | 0.12 | 0.12 | 0.12 | 0.11 | 0.13 | 0.12 |
| AR(2) serial correlation test | 0.19 | 0.23 | 0.22 | 0.25 | 0.24 | 0.24 | 0.25 | 0.75 | 0.19 | 0.25 | 0.21 | 0.25 | 0.24 | 0.24 | 0.24 | 0.68 | 0.19 | 0.25 |

The parentheses are *p*-value. *, **, *** are significance level 10%, 5%, 1%. Instrument variables for period t are utility of the international currency of periods t-3. The null hypothesis of Sargan test is that over-identification is valid. The null hypothesis of AR(1) and AR(2) serial correlation test is that there is no serial correlation.

**Table 5.** Determinants utility of international currency (real interest rate 3.0%).

| | 109 | 110 | 111 | 112 | 113 | 114 | 115 | 116 | 117 | 118 | 119 | 120 | 121 | 122 | 123 | 124 | 125 | 126 |
|---|---|---|---|---|---|---|---|---|---|---|---|---|---|---|---|---|---|---|
| ΔUtility of international currency$_{it-1}$ | −0.03 (0.69) | 0.06 * (0.06) | −0.02 (0.79) | 0.06 ** (0.04) | 0.03 (0.55) | 0.14 (0.24) | 0.05 (0.42) | 0.16 (0.20) | −0.06 (0.57) | 0.05 (0.26) | −0.05 (0.63) | 0.06 (0.17) | 0.29 *** (0.00) | 0.27 ** (0.01) | 0.27 ** (0.01) | 0.30 *** (0.01) | 0.30 *** (0.00) | 0.17 *** (0.00) |
| ΔLiquidity risk premium$_{it}$ | −0.02 (0.42) | −0.03 (0.33) | −0.02 (0.42) | −0.03 (0.33) | −0.03 (0.29) | −0.04 (0.25) | −0.03 (0.28) | −0.04 (0.24) | −0.03 (0.38) | −0.04 (0.29) | −0.03 (0.37) | −0.04 (0.28) | −0.02 (0.41) | −0.02 (0.28) | −0.02 (0.33) | −0.02 (0.32) | −0.02 (0.31) | −0.03 (0.14) |
| ΔMoney stock share$_{it}$ | 0.36 (0.28) | 0.37 (0.28) | 0.38 (0.25) | 0.40 (0.23) | 0.24 (0.67) | 0.33 (0.57) | 0.28 (0.63) | 0.38 (0.51) | 0.20 (0.58) | 0.28 (0.49) | 0.22 (0.53) | 0.32 (0.43) | 0.31 (0.16) | | | | | |
| ΔRelative nominal economic growth$_{it}$ | −0.0003 (0.32) | −0.0006 (0.14) | | | −0.0003 (0.44) | −0.0007 (0.21) | | | −0.0002 (0.31) | −0.0006 (0.12) | | | | −0.0009 (0.20) | | | | |
| ΔRelative real economic growth$_{it}$ | | | 0.0000 (0.99) | 0.0000 (0.90) | | | 0.0000 (1.00) | −0.0001 (0.80) | | | 0.0000 (0.94) | 0.0000 (0.88) | | | −0.0001 (0.79) | | | |
| ΔGDP share$_{it}$ | 0.39 (0.28) | 0.18 (0.33) | 0.39 (0.25) | 0.18 (0.25) | 0.03 (0.94) | −0.19 (0.64) | 0.00 (0.99) | −0.22 (0.59) | 0.27 (0.24) | 0.05 (0.88) | 0.27 (0.24) | 0.04 (0.90) | | | | 0.23 (0.20) | | |
| ΔCapitalization share$_{it}$ | −0.06 (0.83) | −0.20 (0.18) | −0.05 (0.86) | −0.17 (0.25) | 0.06 (0.77) | −0.09 (0.72) | 0.07 (0.72) | −0.07 (0.79) | −0.02 (0.94) | −0.17 (0.26) | −0.02 (0.93) | −0.16 (0.30) | | | | | 0.17 (0.10) | |
| ΔTotal trade share$_{it}$ | −2.45 *** (0.00) | | −2.46 *** (0.00) | | −2.40 *** (0.00) | | −2.39 *** (0.00) | | −2.52 *** (0.00) | | −2.52 *** (0.00) | | | | | | | −1.41 *** (0.00) |
| ΔTotal export share$_{it}$ | | −1.78 ** (0.02) | | −1.81 ** (0.02) | | −1.65 ** (0.01) | | −1.65 ** (0.02) | | −1.80 ** (0.01) | | −1.82 ** (0.02) | | | | | | |
| ΔCapital flow share$_{it}$ | 0.53 (0.28) | 0.52 (0.38) | 0.52 (0.30) | 0.49 (0.42) | 0.65 (0.19) | 0.67 (0.23) | 0.64 (0.20) | 0.65 (0.26) | 0.57 (0.39) | 0.60 (0.41) | 0.56 (0.41) | 0.57 (0.45) | | | | | | |
| ΔSD of Nominal effective exchange rate$_{it}$ | 0.00 (0.53) | 0.00 (0.42) | 0.00 (0.52) | 0.00 (0.40) | | | | | | | | | 0.00 (0.30) | 0.00 (0.30) | 0.00 (0.30) | 0.00 (0.29) | 0.00 (0.27) | 0.00 (0.37) |
| ΔLN Nominal effective exchange rate$_{it}$ | | | | | 0.10 (0.23) | 0.07 (0.38) | 0.11 (0.22) | 0.08 (0.35) | | | | | | | | | | |
| ΔLN Real effective exchange rate$_{it}$ | | | | | | | | | 0.07 (0.32) | 0.04 (0.55) | 0.07 (0.31) | 0.05 (0.50) | | | | | | |
| Sargan test | 0.29 | 0.51 | 0.32 | 0.56 | 0.48 | 0.68 | 0.55 | 0.76 | 0.25 | 0.49 | 0.29 | 0.55 | 0.86 | 0.81 | 0.83 | 0.88 | 0.87 | 0.41 |
| AR(1) serial correlation test | 0.14 | 0.13 | 0.13 | 0.13 | 0.15 | 0.18 | 0.15 | 0.18 | 0.23 | 0.17 | 0.22 | 0.17 | 0.09 * | 0.12 | 0.12 | 0.10 * | 0.11 | 0.14 |
| AR(2) serial correlation test | 0.08 * | 0.15 | 0.08 * | 0.15 | 0.19 | 0.13 | 0.21 | 0.13 | 0.18 | 0.12 | 0.19 | 0.11 | 0.23 | 0.22 | 0.22 | 0.23 | 0.22 | 0.17 |

**Table 5.** *Cont.*

| | 127 | 128 | 129 | 130 | 131 | 132 | 133 | 134 | 135 | 136 | 137 | 138 | 139 | 140 | 141 | 142 | 143 | 144 |
|---|---|---|---|---|---|---|---|---|---|---|---|---|---|---|---|---|---|---|
| ΔUtility of international currency$_{it-1}$ | 0.22 *** | 0.30 *** | 0.31 *** | 0.42 *** | 0.42 *** | 0.43 *** | 0.43 ** | 0.23 | 0.31 * | 0.42 ** | 0.27 *** | 0.41 *** | 0.42 *** | 0.42 *** | 0.43 ** | 0.23 | 0.30 | 0.41 ** |
| | (0.00) | (0.00) | (0.00) | (0.01) | (0.01) | (0.00) | (0.02) | (0.11) | (0.10) | (0.03) | (0.00) | (0.01) | (0.01) | (0.00) | (0.03) | (0.13) | (0.10) | (0.04) |
| ΔLiquidity risk premium$_{it}$ | −0.03 | −0.02 | −0.03 | −0.03 * | −0.03 * | −0.03 * | −0.03 ** | −0.04 ** | −0.04 * | −0.04 * | −0.02 | −0.03 * | −0.03 * | −0.03 * | −0.03 ** | −0.04 * | −0.04 * | −0.03 * |
| | (0.13) | (0.36) | (0.24) | (0.06) | (0.07) | (0.08) | (0.04) | (0.05) | (0.08) | (0.07) | (0.32) | (0.05) | (0.06) | (0.06) | (0.04) | (0.05) | (0.09) | (0.06) |
| ΔMoney stock share$_{it}$ | | | 0.09 | | | | | | | | 0.15 | | | | | | | |
| | | | (0.75) | | | | | | | | (0.55) | | | | | | | |
| ΔRelative nominal economic growth$_{it}$ | | | | −0.0009 | | | | | | | | −0.0009 | | | | | | |
| | | | | (0.27) | | | | | | | | (0.26) | | | | | | |
| ΔRelative real economic growth$_{it}$ | | | | | 0.0000 | | | | | | | | 0.0000 | | | | | |
| | | | | | (0.83) | | | | | | | | (0.84) | | | | | |
| ΔGDP share$_{it}$ | | | | | | −0.18 | | | | | | | | −0.08 | | | | |
| | | | | | | (0.20) | | | | | | | | (0.64) | | | | |
| ΔCapitalization share$_{it}$ | | | | | | | 0.11 | | | | | | | | 0.14 | | | |
| | | | | | | | (0.76) | | | | | | | | (0.72) | | | |
| ΔTotal trade share$_{it}$ | | | | | | | | −1.91 *** | | | | | | | | −1.90 *** | | |
| | | | | | | | | (0.00) | | | | | | | | (0.00) | | |
| ΔTotal export share$_{it}$ | −0.84 | | | | | | | | −1.17 * | | | | | | | | −1.20 ** | |
| | (0.13) | | | | | | | | (0.06) | | | | | | | | (0.05) | |
| ΔCapital flow share$_{it}$ | | 0.47 | | | | | | | | 0.45 *** | | | | | | | | 0.51 *** |
| | | (0.20) | | | | | | | | (0.00) | | | | | | | | (0.00) |
| ΔSD of Nominal effective exchange rate$_{it}$ | 0.00 | 0.00 | | | | | | | | | | | | | | | | |
| | (0.32) | (0.16) | | | | | | | | | | | | | | | | |
| ΔLN Nominal effective exchange rate$_{it}$ | | | 0.06 | 0.08 | 0.10 | 0.13 | 0.09 | 0.20 * | 0.15 | 0.06 | | | | | | | | |
| | | | (0.63) | (0.31) | (0.22) | (0.24) | (0.32) | (0.08) | (0.11) | (0.45) | | | | | | | | |
| ΔLN Real effective exchange rate$_{it}$ | | | | | | | | | | | 0.03 | 0.07 | 0.08 | 0.09 | 0.07 | 0.19 * | 0.14 | 0.03 |
| | | | | | | | | | | | (0.81) | (0.41) | (0.30) | (0.40) | (0.44) | (0.10) | (0.13) | (0.67) |
| Sargan test | 0.73 | 0.82 | 0.93 | 0.99 | 0.99 | 0.99 | 0.99 | 0.94 | 0.98 | 0.97 | 0.86 | 0.98 | 0.99 | 0.99 | 0.99 | 0.93 | 0.97 | 0.96 |
| AR(1) serial correlation test | 0.13 | 0.10 | 0.11 | 0.11 | 0.11 | 0.12 | 0.11 | 0.11 | 0.14 | 0.10 | 0.10 | 0.11 | 0.11 | 0.11 | 0.11 | 0.12 | 0.15 | 0.11 |
| AR(2) serial correlation test | 0.19 | 0.22 | 0.21 | 0.25 | 0.25 | 0.24 | 0.25 | 0.68 | 0.20 | 0.25 | 0.21 | 0.25 | 0.24 | 0.24 | 0.25 | 0.60 | 0.19 | 0.24 |

The parentheses are *p*-value. *, **, *** are significance level 10%, 5%, 1%. Instrument variables for period t are utility of the international currency of periods t-3. The null hypothesis of Sargan test is that over-identification is valid. The null hypothesis of AR(1) and AR(2) serial correlation test is that there is no serial correlation.

## 7. Conclusions

In this paper, we investigated what determines utility of the international currencies among the current major currencies which include the US dollar, the euro, the Japanese yen, and the British pound. We used a dynamic panel data model to analyze the issue with GMM. We focused on effects of liquidity shortage in terms of an international currency on utility of the international currencies as well as inertia of the US dollar as the key currency. We made empirical analysis of whether liquidity risk premium in an international currency as well as other possible determinant factors affect utility of the relevant international currency.

We obtained the following results from the empirical analysis. Firstly, change in utility of the international currency in the previous period has significantly a positive effect on the change of utility of the international currency in the current period. This suggests that utility of the international currency tends to fluctuate in the same direction as the change in the previous term. For example, if the utility of the international currency decreases, we assumed that the currency is less likely to be used than in the previous period, which will continue in the next period. Secondly, the change of liquidity risk premium has significantly a negative effect on the change of utility of the currency. This suggests that liquidity shortage reduce the utility of the international currency. Thirdly, the change of capital flow share has significantly a positive effect on the change of utility of the international currency. This suggests that increase in economic scale may increase utility of the international currency.

We mention policy implications from the empirical results. As mentioned above, liquidity risk premium and capital flow had influence on utility of the international currency. If the monetary authorities try to internationalize their home currencies, it is necessary to focus on these variables. It is considered that utility of the international currency will increase by conducting policies that increase liquidity of the currency or increase international capital flows. It might be possible to push internationalization of the home currencies through this increase of utility of the international currencies.

Moreover, in this analysis, we found a strong relationship between utility of the international currency and liquidity of the international currency. We should also deeply analyze relationship between the two variables. In this analysis, as we supposed that utility of the international currency is dependent variable, changes in liquidity of the international currency have influenced changes in the utility of the international currency. However, it is necessary for us to suppose causality between the liquidity of the international currency and its utility in the opposite direction. If utility of an international currency declines, it may reduce liquidity of the international currency through reduction in convenience of the currency.

We used liquidity risk premium as a variable for liquidity of the international currency in this paper. There are such other variables as bid-ask spreads that represent liquidity of the international currency. Moreover, it is capital adequacy of financial institution that could affect liquidity of the international currency. It is necessary to conduct robustness tests for the analysis result by conducting any analysis using the variables in the future. Furthermore, we used ARIMA model using CPI for the expected inflation rate. Other expected inflation rate data include TIPS data and survey data, which may reflect the reality. Robustness check using these data as future work is also necessary.

In addition, we have further study regarding what factors are important to help an emerging new international currency which includes the Chinese yuan. In recent years, the Chinese government has been promoting the Chinese yuan to be internationalized while the IMF has added it into major international currencies that is component currencies of the Special Drawing Rights (SDR). It is important for us to investigate how a local currency can emerge as an international currency and, in turn, make it into a key currency in the current international monetary system where we do not as a rule set any currencies as a key currency.

**Author Contributions:** Conceptualization, E.O.; methodology, E.O.; formal analysis, M.M.; investigation, M.M.; writing—original draft preparation, E.O. and M.M.; writing—review and editing, E.O. and M.M.; visualization, E.O. and M.M.; supervision, E.O.; project administration, E.O.

**Funding:** This research received no external funding.

**Acknowledgments:** This study is conducted as a part of the project "Exchange Rate and International Currency" undertaken at the Research Institute of Economy, Trade and Industry (RIETI). The authors are grateful to the three anonymous referees, Keiichiro Kobayashi, Etsuro Shioji, Shinichi Fukuda, Jorg Mayer, Stephan Gerlach and other participants in the RIETI research seminar, a research meeting of the Ministry of Finance, and conferences of the Asia-Pacific Economic Association and Japan Economic Network for their useful comments and suggestions.

**Conflicts of Interest:** The authors declare no conflict of interest.

## Appendix A. Derivation of Equation (1)

We base on a Sidrauski (1967)-type of money-in-the-utility model in which real balances of money as well as consumption are supposed as explanatory variables in a utility function. According to Ogawa and Sasaki (1998), we extend the money-in-the-utility model to a dynamic one with parallel international currencies[5]. We suppose that the international currencies are held by private economic agents in a third country. For simplicity, we suppose that two monetary authorities supply international currencies.

For convenience, we suppose that it is both the monetary authorities in the Country $i$ and other countries $O$ that supply their international currencies. The monetary authorities in Country $i$ supply currency $i$ while the monetary authorities in other countries $O$ supply their own currencies $O$. The private sector in the third country $A$ is able to use both the currencies $i$ and $O$ as international currencies in international economic transactions. The monetary authorities in country $A$ adopt a flexible exchange rate system.

We suppose a situation that bonds in currencies $i$ and $O$ are available to the private sector in country $A$ and that no bonds denominated in currency $A$ are issued in country $A$. We assume perfect capital mobility and perfect substitution for the bonds of different currencies. Moreover, we assume that the private sector has perfect foresight. Thus, uncovered interest parity holds in the model. On one hand, we assume perfect flexible prices and a law of one price. Thus, the purchasing power parity always holds in the model. For simplicity, we assume that its rate of time preference is constant over time and is equal to a real interest rate. Given the assumptions, the real interest rate is constant over time. Real interest rates in all countries are equal to each other by both the uncovered interest parity and purchasing power parity.

The private sector in country $A$ holds home currency $A$, international currencies $i$ and $O$, and bonds in currencies $i$ and $O$. Instantaneous budget constraints for the private sector are represented in real terms:

$$\dot{w}_t^p = \bar{r}w_t^p + y_t - c_t - \tau_t - i_t^A m_t^A - i_t^i m_t^i - i_t^O m_t^O \tag{A1a}$$

$$w_t^p = b_t^i + b_t^O + m_t^A + m_t^i + m_t^O \tag{A1b}$$

where $y$: real gross domestic products, $\tau$: real taxes, $c$: real consumption, $i^j$: nominal interest rate in currency $j$ ($j = A, i, O$), $w^p$: real balance of financial assets held by the private sector, $m^j$: real balance of home currency $j(j = A, i, O)$ held by the private sector, $b^j$: real balance of bond in currency $j$ ($j = i, O$) held by the private sector, $\bar{r}$: real interest rate. A dot over variables implies a change in the relevant variables. We assume no-Ponzi game conditions for the real balance of financial assets held by the private sector ($w^p$).

$$\lim_{t \to \infty} w_t^p e^{-\bar{r}t} \geq 0 \tag{A2}$$

We assume that the private sector maximizes its utility over an infinite horizon subject to budget constraints (A1a) and (A1b). We specify a Cobb-Douglas type of instantaneous utility function:

$$\int_0^\infty U(c_t, m_t^A, m_t^i, m_t^O)e^{-\delta t}dt \tag{A3a}$$

---

[5]　See Ogawa and Muto (2017a) for the detailed derivation.

$$U(c_t, m_t^A, m_t^i, m_t^O) \equiv \frac{\left[ c_t^\alpha \left\{ m_t^{A\beta} \left( m_t^{i\gamma} m_t^{O1-\gamma} \right)^{1-\beta} \right\}^{1-\alpha} \right]^{1-R}}{1-R} \tag{A3b}$$

$$0 < \alpha < 1,\ 0 < \beta < 1,\ 0 < \gamma < 1,\ 0 < R < 1$$

where $\delta$: rate of time preference, $R$: reciprocal of instantaneous elasticity of substitution between intertemporal consumption $\sigma$: $\sigma \equiv -\frac{U_c}{U_{cc} c_t}$

We assume that the public sector in country $A$ holds only bonds in currencies $i$ and $O$. Instantaneous budget constraints for the public sector are represented in real terms:

$$\dot{f}_t = \bar{r} f_t + \tau_t + \mu_t^A m_t^A - g_t \tag{A4a}$$

$$f_t \equiv f_t^i + f_t^O \tag{A4b}$$

where $g$: real government expenditures, $f$: foreign assets held by the public sector, $\mu^A$: growth rate of currency $A$. We assume no-Ponzi game conditions for foreign assets held by the public sector.

$$\lim_{t \to \infty} f_t e^{-\bar{r} t} \geq 0 \tag{A5}$$

A stock of foreign exchange reserves held by the monetary authorities should be unchanged under a flexible exchange rate system because the monetary authorities will not intervene in foreign exchange markets ($f_t = \bar{f}$). Also, they are able to control nominal money supply. Here, we assume that they increase the nominal money supply at a constant growth rate $\mu^A$.

Thus we obtain an instantaneous budget constraint equation for the public sector under a flexible exchange rate system:

$$g_t - \tau_t = \bar{r} \bar{f} + \bar{\mu}^A m_t^A \tag{A6}$$

From the instantaneous budget constraint equations for the private sector and the public sector Equations (A1a) and (A6), we derive an instantaneous budget constraint equation for the whole economy of country A under a flexible exchange rate system:

$$\dot{b}_t^i + \dot{b}_t^O + \dot{m}_t^i + \dot{m}_t^O = \bar{r}(b_t^i + b_t^O + m_t^i + m_t^i + \bar{f}) + y_t - c_t - g_t - i_t^i m_t^i - i_t^O m_t^O \tag{A7}$$

The private sector maximizes its utility Functions (A3a) and (A3b) subject to budget constraint Equation (A7). We assume that the private sector has perfect foresight that economic variables do not diverge to infinity but converge to equilibrium values along a saddle path to rule out a possibility of multiplicity of equilibria in the model.

From the first-order conditions for maximization, we derive optimal real balances of international currencies:

$$m_t^i = \frac{(1-\alpha)(1-\beta)\gamma}{\alpha} \frac{\bar{c}}{i_t^i} = \frac{(1-\alpha)(1-\beta)\gamma}{\alpha} \frac{\bar{c}}{i_t^i + \bar{r}} \tag{A8a}$$

$$m_t^O = \frac{(1-\alpha)(1-\beta)(1-\gamma)}{\alpha} \frac{\bar{c}}{i_t^O} = \frac{(1-\alpha)(1-\beta)(1-\gamma)}{\alpha} \frac{\bar{c}}{i_t^O + \bar{r}} \tag{A8b}$$

where $\pi_t^j$: inflation rate of currency $j (j = i, O)$,

$$\bar{c} = \bar{r} \left\{ a_0 + \int_0^\infty y_t e^{-\bar{r} t} dt - \int_0^\infty g_t e^{-\bar{r} t} dt - \int_0^\infty (i_t^i m_t^i - i_t^O m_t^O) e^{-\bar{r} t} dt \right\}$$

From Equations (A8a) and (A8b), an optimal share $\phi$ of $i$ is derived:

$$\phi_t \equiv \frac{m_t^i}{m_t^i + m_t^O} = \frac{1}{1 + \frac{1-\gamma}{\gamma}\frac{i_t^i}{i_t^O}} = \frac{1}{1 + \frac{1-\gamma}{\gamma}\frac{\pi_t^i + \bar{r}}{\pi_t^O + \bar{r}}} \tag{A9}$$

Equation (A9) implies that the optimal share of $i$ depends on both the inflation or depreciation rates of the international currencies ($\pi^i$ and $\pi^O$) and a parameter $\gamma$ in the utility function Equation (A3b). From Equation (A9), the parameter $\gamma_t^i$ is derived:

$$\gamma_t^i = \frac{1}{1 + \left(\frac{1}{\phi_t^i} - 1\right)\frac{\pi_t^O + \bar{r}}{\pi_t^i + \bar{r}}} \tag{A10}$$

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
