# Peer review of "What Determines Utility of International Currencies?"

_jrfm, doi:10.3390/jrfm12010010_

Reviewer 1 Report

This paper tries to study an interesting question on how to empirically evaluate the utility for holding a foreign currency based on a model motivated by an MIU model. I have the following concerns:

(1) Theoretical model: the gamma is solved from equation (A9). However, from A9 we can see that, the share of foreign currency, phi, is endogenous while gamma and real interest rate are exogenous. If I am correct, it does not make any sense to me for the calculation of gamma. If the author consider that gamma is endogenous, then it makes more sense to solve the model by expressing the endogenous variables in terms of exogenous variables. Then calculate the value of the endogenous variable according to the value of exogenous variable.

(2) Theoretical model again: the author tries to derive the utility (gamma) according to a non-stochastic MIU model. This model describes the long-term behavior of the economy. But in the empirical analysis, the author use quarterly data for only 10 years. I do not believe that it is consistent between the theoretical model and the empirical model.

(3) real interest rate: the authors sample are from 2006 and 2016 and then the author set the real interest rate as some number between 1.5 and 3.  Nobody believe that the real interest rate is so high for advanced economies in this period. Even nominal interest rate was around 0 for most of the countries. This makes the empirical analysis of little practical use. In addition, the natural interest rate declined for advanced economies. (See Holston, Laubach and Williams(2017, JIE)

(4) I do not see much difference between the inflation expectation and the actual inflation. But in reality, can people perfectly forecast the inflation? It might be better for the author to use survey data or TIPS inflation instead of using an overfitted statistical model to capture the inflation expectation.

(5) The writing needs to be improved. There are some grammatical issues to be corrected.

Author Response

(1) Theoretical model: the gamma is solved from equation (A9). However, from A9 we can see that, the share of foreign currency, phi, is endogenous while gamma and real interest rate are exogenous. If I am correct, it does not make any sense to me for the calculation of gamma. If the author consider that gamma is endogenous, then it makes more sense to solve the model by expressing the endogenous variables in terms of exogenous variables. Then calculate the value of the endogenous variable according to the value of exogenous variable.

Response: An important objective of the paper is to analyze what factors influence utility of the currency gamma during the analytical period. For the purpose, we suppose that gamma might change over time though it seems to be stable as a exogenous. From the above, we calculated the gamma at each period from share of holdings of international currency phi, expected inflation rate and real interest rates. At footnote 1 on page 3, we have revised that we suppose that gamma might change over time though it seems to be stable as a exogenous.

(2) Theoretical model again: the author tries to derive the utility (gamma) according to a non-stochastic MIU model. This model describes the long-term behavior of the economy. But in the empirical analysis, the author use quarterly data for only 10 years. I do not believe that it is consistent between the theoretical model and the empirical model.

Response: As mentioned in lines 291 to 292 on page 11, the Japanese yen liquidity risk premium data has constraint. However, in this paper, We focused on effects of liquidity shortage in terms of an international currency on utility of the international currencies. Therefore, we could not increase sample size by excluding liquidity risk premium data. In addition, even if Japan were excluded from analysis, the sample size almost did not improve due to constraint of euro data. Therefore, We used data from 2006Q3 to 2017Q4 although the sample size is small.

(3) real interest rate: the authors sample are from 2006 and 2016 and then the author set the real interest rate as some number between 1.5 and 3. Nobody believe that the real interest rate is so high for advanced economies in this period. Even nominal interest rate was around 0 for most of the countries. This makes the empirical analysis of little practical use. In addition, the natural interest rate declined for advanced economies. (See Holston, Laubach and Williams(2017, JIE)

Response: It is questionable whether it is adequate to use a sum of the real interest rate and an expected inflation rate in a zero-bound interest rate period after the global financial crisis. We originally used nominal interest rates to estimate utility of the currency. However, we had to use a sum of the real interest rate and an expected inflation rate as a proxy of nominal interest rate because the nominal interest rate had strange movements especially during the global financial crisis when European financial institutes faced US dollar liquidity shortage. As mentioned in footnote 2 on page 4, we have used an arithmetic average of real economic growth rates as a reference value for real interest rates.

(4) I do not see much difference between the inflation expectation and the actual inflation. But in reality, can people perfectly forecast the inflation? It might be better for the author to use survey data or TIPS inflation instead of using an overfitted statistical model to capture the inflation expectation.

Response: We could not obtain expected inflation rate data of one quarter ahead from other data. Therefore, we estimated in ARIMA model using CPI data. However, we believe that it is also necessary to perform robustness checks using expected inflation rate of survey data and TIPS data. We revised the above mentioned about expected inflation rate in footnote 3 on page 4 and lines 529 to 531 on page 26.

    (5) The writing needs to be improved. There are some grammatical issues to be corrected.

Response: We did improve the article to correct the grammatical issues.

Reviewer 2 Report

This paper investigates what determines utility of major international cuurencies by employing a dynamic panel and GMM methodology and a utility function. Findings indicate that liquidity risk premium and capital flow did have an impact on utility of currencies. Liquidity shortgae is found to have exerted important effects in international currencies.

The overall presentation and analysis of this paper is very good, apart from the literature review section that needs to be strengthened. Econometric findings are interesting as well as the topic investigated. Therefore, I believe that this paper should be accepted under minor revision.

Author Response

This paper investigates what determines utility of major international currencies by employing a dynamic panel and GMM methodology and a utility function. Findings indicate that liquidity risk premium and capital flow did have an impact on utility of currencies. Liquidity shortage is found to have exerted important effects in international currencies.

The overall presentation and analysis of this paper is very good, apart from the literature review section that needs to be strengthened. Econometric findings are interesting as well as the topic investigated. Therefore, I believe that this paper should be accepted under minor revision.

Response: We revised the literature review section in lines 111 to 114 and 117 to 126 on page 3.

Reviewer 3 Report

While this paper has sound econometric quality, the theoretical model is questionable as is the research question. The paper does not address what the utility of a currency means and why we should be interested in it. Within a C-D framework the composition of the foreign currencies is as it rightly states the shares of the currency in use by the portfolio manager of the home country. Of the determinants, liquidity is as the authors state, jointly determined. The paper lays down a maker for further work, which is good. Then this paper should be clear that it is preliminary and to be improved in future work. 

Typo in line 586 Greek psi should Greek phi.

Author Response

(1) While this paper has sound econometric quality, the theoretical model is questionable as is the research question. The paper does not address what the utility of a currency means and why we should be interested in it. Within a C-D framework the composition of the foreign currencies is as it rightly states the shares of the currency in use by the portfolio manager of the home country. Of the determinants, liquidity is as the authors state, jointly determined. The paper lays down a maker for further work, which is good. Then this paper should be clear that it is preliminary and to be improved in future work.

Response: We have revised lines 52 to 57 and lines 65 to 67 on page 2 about what utility of a currency means and why we should be interested in it. In addition, we have revised lines 532 to 538 on page 26 about future work to be improved.

     (2) Typo in line 586 Greek psi should Greek phi.

Response: We have revised as the Greek phi fonts were different.
